# Modeling the Human Visual System: Comparative Insights from Response-Optimized and Task-Optimized Vision Models, Language Models, and Different Readout Mechanisms

**Shreya Saha (ssaha@ucsd.edu)**
Electrical and Computer Engineering
University of California, San Diego

**Ishaan Chadha (ichadha@ucsd.edu)**
Halıcıoğlu Data Science Institute
University of California, San Diego

**Meenakshi Khosla (mkhosla@ucsd.edu)**
Department of Cognitive Science, Department of Computer Science and Engineering
University of California, San Diego

## Abstract

**Over the past decade, predictive modeling of neural responses in the primate visual system has advanced significantly, driven by diverse deep neural network approaches. These include models optimized for visual recognition, methods that align visual and language information, models trained directly on brain data, and representations from large language models (LLMs). Additionally, various readout mechanisms have been developed to map network activations to neural responses. Despite this progress, it remains unclear which approach performs best across different regions of the visual hierarchy. In this study, we systematically compare these methods for modeling the human visual system and propose novel strategies to enhance response predictions. We demonstrate that the choice of readout mechanism significantly impacts prediction accuracy and introduce a biologically grounded readout that dynamically adjusts receptive fields based on image content and learns geometric invariances of voxel responses directly from data. This novel readout outperforms factorized methods by 3-23% and standard ridge regression by 7-53%, setting a new benchmark for neural response prediction. Our findings reveal distinct modeling advantages across the visual hierarchy: response-optimized models with visual inputs excel in early to mid-level visual areas, while embeddings from LLMs—leveraging detailed contextual descriptions of images—and task-optimized models pretrained on large vision datasets provide the best fit for higher visual regions. Through comparative analysis, we identify three functionally distinct regions in the visual cortex: one sensitive to perceptual features not captured by linguistic descriptions, another attuned to fine-grained visual details encoding semantic information, and a third responsive to abstract, global meanings aligned with linguistic content. Together, these findings offer key insights into building more precise models of the visual system.**

**Keywords:** Neuro AI, vision, deep neural networks, Neural Response Modeling, fMRI encoding, Readout Mechanisms, Vision Language Alignment

## Introduction and Related Work

Building accurate predictive models of the visual system has been a longstanding goal in neuroscience. Early approaches primarily relied on handcrafted features, such as Gabor filters, curvature models, and motion energy models, to predict responses in early to mid-level visual areas (Hubel & Wiesel, 1962; Livingstone & Hubel, 1984; Albrecht & Hamilton, 1982; Gallant et al., 1993; Hubel & Wiesel, 1968; Desimone et al., 1984; Tanaka et al., 1991; Pasupathy & Connor, 2002; Yue et al., 2020; Yang et al., 2023; Pasupathy & Connor, 1999; Tsunoda et al., 2001; Rust & DiCarlo, 2010; Brincat & Connor, 2004; Zeki, 1973; Pasupathy & Connor, 2001; Moran & Desimone, 1985; Kobatake & Tanaka, 1994; Kriegeskorte et al., 2008; Kobatake et al., 1998; Miyashita, 1988). Similarly, word-based descriptions were often used to model responses in higher-level visual regions (Huth et al., 2012). These models provided interpretability, as the features they employed were well understood and linked to specific visual computations. However, they lacked quantitative precision in their ability to predict neural responses. [1]

The advent of deep convolutional neural networks (DCNNs) marked a significant improvement in predictive accuracy across the visual system (Yamins et al., 2014; Abdelhack & Kamitani, 2018; Wen et al., 2018; Horikawa & Kamitani, 2017; Eickenberg et al., 2017; Güçlü & Van Gerven, 2015; Cichy et al., 2016; Khaligh-Razavi & Kriegeskorte, 2014; Schrimpf et al., 2020; Storrs et al., 2021; Safarani et al., 2021; Schwartz et al., 2019; Seeliger et al., 2021; Shen et al., n.d.). DCNNs trained on image categorization tasks emerged as the first class of models capable of capturing neural activity in the primate visual cortex with a reasonable degree of fidelity. This success spurred a wave of model-brain comparisons, wherein

---

[1] Code can be found at - https://github.com/NeuroML-Lab/Visual-Stream-Modeling

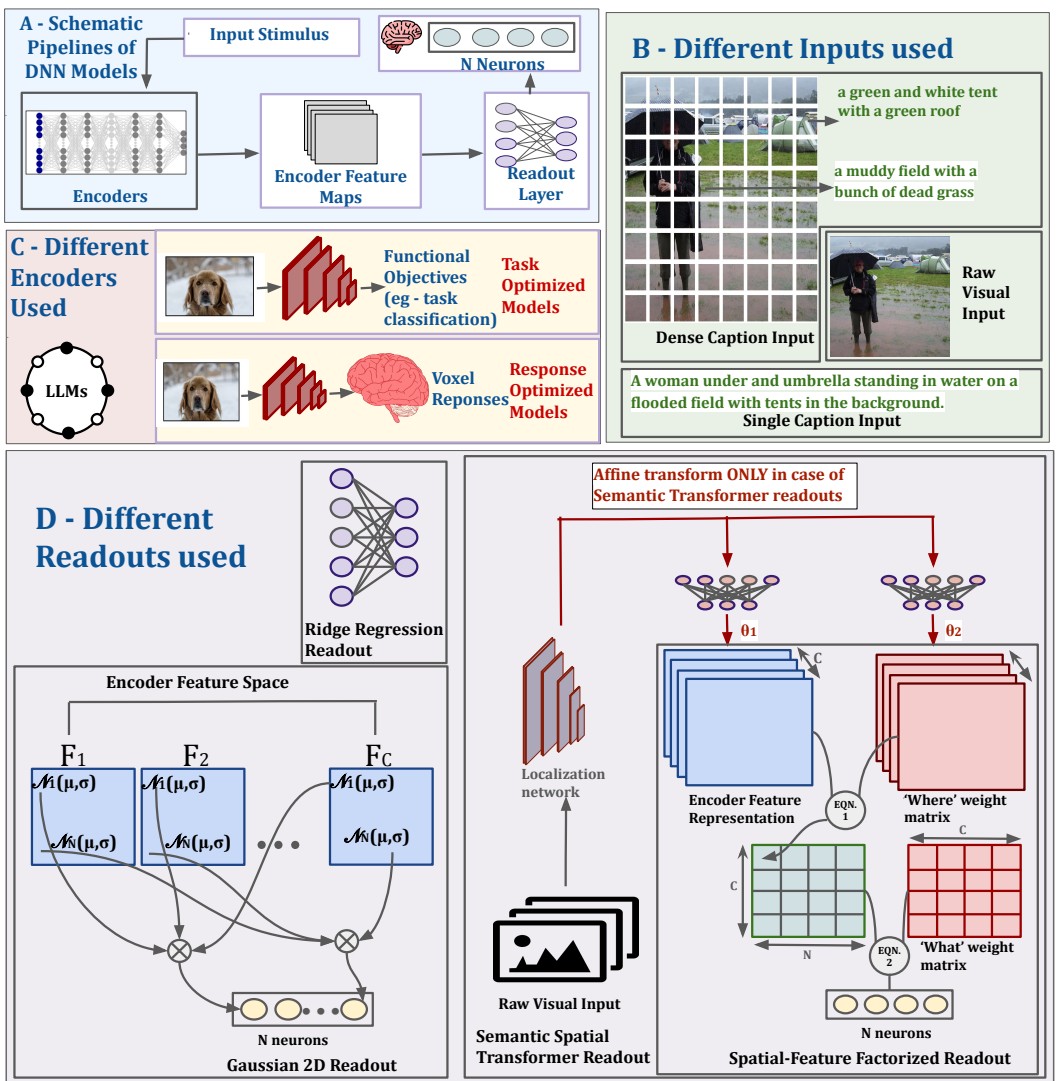

Figure 1: (A) High-level schematic of the key components analyzed in this study. (B) Various stimuli used to model the visual cortex. (C) Different encoder backbones employed in the study. (D) Readout mechanisms (Linear, Gaussian, Factorized, and Semantic Spatial Transformer) that map ANN encoder representations to neuronal or voxel responses.

variations in input data, architecture, and learning objectives were explored to identify the most predictive models of brain responses in both non-human primates and humans.

More recently, models trained using multimodal contrastive learning approaches, such as CLIP, or image-caption embeddings from large language models (LLMs), have shown promise in predicting neural responses in the visual cortex (Tang et al., 2024; Wang et al., 2022; Doerig et al., 2024). These findings suggest that visual brain responses may encode some linguistically learned structure or semantics. In parallel, another class of models, optimized specifically for neural response prediction (Khosla & Wehbe, 2022; Khosla et al., 2022; Federer et al., 2020; Dapello et al., 2022; St-Yves et al., 2023) — either trained from scratch or fine-tuned to better align with primate visual representations—has achieved impressive predictive accuracy, particularly with the availability

of large-scale neural datasets (Allen et al., 2022).

Given the broad range of modeling approaches applied to different regions of the visual cortex, a critical question remains: which approach offers the most quantitatively precise predictions of neural responses across the various areas of the human visual system? This challenge underscores the need for systematic comparisons to determine the optimal models for different visual processing stages. While some recent studies have made strides in conducting large-scale comparative analyses, they tend to focus primarily on specific pre-selected visual regions and largely compare different task-optimized vision networks (Conwell, Prince, Kay, et al., 2022). A more comprehensive comparison is needed to evaluate a broader set of approaches, including models based on response optimization and embeddings from language models trained on vision-aligned tasks or pure language data.

An equally pressing issue concerns the readout mechanism by which models' internal representations are mapped onto neural responses Ivanova et al. (2022). The predominant readout in primate studies is the fully-connected affine readout, often used in regularized linear regression models. However, these linear ridge regression readouts require numerous parameters, especially in high-dimensional spaces, leading to significant computational and memory demands. To mitigate this, more efficient methods have been developed, such as factorized linear readouts by (Klindt et al., 2017), that decouple spatial from feature selectivity, reducing overhead and improving prediction accuracy. The Gaussian2D readout (Lurz et al., 2020) further enhances parameter efficiency by learning spatial readout locations using a bivariate Gaussian distribution informed by anatomical retinotopy. However, it is still unclear which readout approach provides the best predictive accuracy across different cortical areas.

Determining the most effective model—and the most suitable readout—for each region of the visual cortex is vital. Accurate models provide a powerful platform for *in silico* experimentation, enabling researchers to test hypotheses that may be impractical to probe *in vivo*. They also inform experimental design and facilitate precise neural population control (Walker et al., 2019; Bashivan et al., 2019). In this way, achieving high predictive accuracy is foundational for both practical applications and deeper theoretical insights into visual processing.

In this paper, we bridge these gaps by systematically comparing a broad array of models—along with diverse readout methods—to identify the most accurate approach for each region of the human visual cortex. Specifically, we make the following key contributions:

1. **Comprehensive analysis of different neural network models and readouts:** We systematically compare an extensive set of neural network models spanning vision-only, vision-language and language-only paradigms. Additionally, we explore different readout mechanisms and examine which models perform better in specific brain regions, while highlighting the unique advantages each provides.
2. **Introduction of a novel readout:** We introduce a novel biologically-grounded readout method which delivers significant improvements in accuracy, outperforming factorized methods by 3-23% and standard ridge regression (the de facto choice in many studies) by 7-53% .
3. **Identification of brain regions sensitive to perceptual and semantic information:** Through large-scale comparative analysis of models across various visual regions, we identify three distinct regions in the human visual cortex that respond primarily to (a) low-level perceptual characteristics of the input, (b) localized visual semantics aligned with linguistic descriptions, and (c) global semantic interpretations of the input, also aligned with language.

## Methods

### Encoders

**Task-optimized Models**   We use encoders from pre-trained models like AlexNet (Krizhevsky et al., 2012) and ResNet (He et al., 2016), originally trained for object classification on the large-scale ImageNet dataset (Deng et al., 2009). The weights of their intermediate layers are frozen, and only the readout layers (described later) are trained. Prior research shows that early layers of neural networks align with lower visual cortex regions, while later layers correspond to higher regions (Khaligh-Razavi & Kriegeskorte, 2014; Güçlü & Van Gerven, 2015; Cichy et al., 2016; Eickenberg et al., 2017; Horikawa & Kamitani, 2017; Wen et al., 2018; Abdelhack & Kamitani, 2018; Yamins et al., 2014). Thus, we experimented with all layers of task-optimized networks. For fair comparison, we selected the best-performing layers for each cortical region (see Appendix Table A1 and summary in Table 1).

**Response-optimized Models**   Task-optimized models often rely heavily on a priori hypotheses, which may be biased towards pre-existing conclusions, limiting novel discoveries. Further, these networks are typically optimized for specific tasks, such as object classification, which may not capture the full range of visual processing in the cortex. Recently, (Khosla & Wehbe, 2022) showed that training neural networks from scratch with stimulus images and fMRI data from the NSD dataset (Allen et al., 2022) can achieve accuracy comparable to state-of-the-art task-optimized models. By directly optimizing for neural responses, these models are free to learn representations that are more closely aligned with the underlying neural computations, unencumbered by the biases inherent in task-driven models. This flexibility can enable response-optimized models to uncover richer, more generalizable representations that better reflect the diversity of neural activation patterns across brain regions.

We leverage the same architecture for response-optimized models as prior work (Khosla & Wehbe, 2022), which consists of a convolutional neural network (CNN) core that transforms raw input data into feature spaces characteristic of different brain regions, followed by a readout layer that maps these features to fMRI voxel responses. The core contains four convolutional blocks, where each convolutional block includes two convolutional layers, followed by internal batch normalization, nonlinear ReLU activations, and an anti-aliased average pooling operation. To ensure equivariance under all isometries, we use E(2)-Equivariant Steerable Convolution layers (Weiler & Cesa, 2019). Further analysis on the importance of network architecture for Response-optimized models can be found in Appendix section Comparing different architectures for Task and Response Optimized models and Table A6.

**Language Models**   - Recent studies show that higher visual regions converge toward representational formats similar to large language model (LLM) embeddings of scene descriptions. (Doerig et al., 2024) used MPNET (Song et al., 2020) to encode image captions and map them to fMRI re-

sponses via ridge regression, finding it effectively modeled higher visual areas despite being trained on language inputs alone. In contrast, (Tang et al., 2024) and (Wang et al., 2022) used multimodal models like CLIP (Radford et al., 2021) and BridgeTower (Yang et al., 2023), showing that CLIP outperforms vision-only models in capturing higher visual regions, attributing this to language feedback. These motivated us to evaluate language models relative to vision-only response-optimized and task-optimized models as detailed below (More detailed comparison on CLIP and MPNET embeddings and additional results with GPT2-XL Brown et al. (2020) can be found in Appendix section Unimodal versus multimodal embeddings in language models and Table A3) -

1. **Single Caption** - Images in the NSD dataset are sourced from MS COCO (Lin et al., 2014) and annotated by 4-5 human annotators. We encode these captions using CLIP or MPNET, average the encodings, and input them into a linear regressor to map them to fMRI voxel responses. Since the captions describe the image as a whole without offering spatial details (i.e., fine-grained delineations of features at different locations), we only use the ridge linear readout for single caption inputs.

2. **Dense Caption** - An image of size $424 * 424$ is divided into grids of size $53 * 53$. For each grid, a caption is generated using GPT-2, which is then encoded by either CLIP or MP-NET. Thus an image of shape $3 * 424 * 424$ is transformed into a feature representation $N * 8 * 8$, where N is the size of the embedding produced by CLIP or MPNET. The dense-caption language encoders further process these feature maps through a single convolutional block (as described earlier for the response-optimized vision encoders) before passing them to the readout model. For additional technical details—including an in-depth examination of whether dense-caption improvements stem from spatial subdivision or increased semantic detail, as well as experiments comparing alternative single-caption approaches—please refer to the Appendix section The Necessity of Spatial Subdivision in Dense Captioning for Effective Visual Cortex Modeling, Table A7 and Figure A5.

### Readouts

The encoders discussed above are paired with a readout model (Figure 1) that maps the encoder feature representations to voxel fMRI responses from various regions of the visual cortex.

**Linear Readout**  This approach uses a ridge regression model to map encoder features directly to voxel responses. Let $n$ be the total number of voxels in the measured brain region. For a given stimulus $i$, the predicted voxel response vector $\hat{\mathbf{Y}}_i \in \mathbb{R}^n$ is computed as $\hat{\mathbf{Y}}_i = \mathbf{W}\mathbf{E}_i$, where $\mathbf{E}_i \in \mathbb{R}^e$ is the flattened encoder feature representation and $\mathbf{W} \in \mathbb{R}^{n \times e}$ is the weight matrix. These weights are learned by minimizing the ridge regression objective: $\min_{\mathbf{W}} \|\mathbf{Y} - \mathbf{W}\mathbf{E}\|_F^2 + \lambda\|\mathbf{W}\|_F^2$, where $\mathbf{Y}$ is the matrix of true voxel responses, $\mathbf{E}$ is the corresponding matrix of encoder features, $\|\cdot\|_F$ denotes the Frobe-

nius norm, and $\lambda$ is the regularization parameter. We select the optimal $\lambda$ via cross-validation.

**Spatial-Feature Factorized Linear Readout**  factorizes the linear readout model into spatial (the portion of the input space a voxel is sensitive to) and feature (the specific features of the input space a voxel responds to) dimensions, as described in (Klindt et al., 2017). By separating spatial (where) and feature (what) dimensions, the model mirrors the known structure of neural receptive fields in the brain, where neurons exhibit sensitivity to specific spatial locations and particular feature types. This approach not only significantly reduces the number of parameters but also aligns more closely with the known characteristics of neural responses.

$$\hat{Y}_{c,n} = \sum_{w=1}^{W} \sum_{h=1}^{H} \mathbf{E}_{c,w,h} \mathbf{S}_{n,w,h}, \quad \hat{Y}_n = \sum_{c=1}^{C} \hat{Y}_{c,n} \mathbf{F}_{n,c}. \quad (1)$$

Here, $\hat{Y}_n$ represents the predicted response for voxel $n$, and $\mathbf{E} \in \mathbb{R}^{C \times W \times H}$ is the encoder feature map (the "what"). The spatial weights $\mathbf{S} \in \mathbb{R}^{N \times W \times H}$ specify the receptive field (the "where") for each of the $N$ voxels, while the feature weights $\mathbf{F} \in \mathbb{R}^{N \times C}$ determine each voxel's sensitivity to the $C$ feature channels. $W$ and $H$ denote the spatial dimensions of the encoder feature map.

**Gaussian 2D Readout**  This readout models each voxel's spatial sensitivity as a 2D Gaussian in the encoder feature space (Lurz et al., 2020). Specifically, each voxel $n$ is associated with a bivariate Gaussian distribution $G_n(x,y) \sim \mathcal{N}(\mu_n, \Sigma_n)$, whose mean $\mu_n$ represents the voxel's preferred location (receptive field center), and whose covariance $\Sigma_n$ defines the size, shape, and orientation of the receptive field along the $x$ and $y$ axes. The same spatial Gaussian is applied uniformly across all feature channels, indicating a shared positional sensitivity for each voxel.

To compute the response $\hat{Y}_n$ of voxel $n$, we first bilinearly interpolate the feature values $\mathbf{V}_c(x,y)$ from channel $c$ of the encoder feature map $\mathbf{E} \in \mathbb{R}^{C \times W \times H}$ at spatial coordinates $(x,y)$, weighted by the Gaussian distribution $G_n(x,y)$. We then multiply these interpolated values by the learned channel-specific weights $\mathbf{W}_{nc}$ and sum over channels: $\hat{Y}_n = \sum_{c=1}^{C} \mathbf{W}_{nc} \mathbf{V}_c(x,y)$. Here, $\mathbf{W}_{nc}$ determines the contribution of channel $c$ to voxel $n$, and the interpolated feature $\mathbf{V}_c(x,y)$ depends on the Gaussian weighting specified by $G_n(x,y)$. By incorporating spatial information in this way, the Gaussian 2D readout captures the spatial sensitivity of each voxel with fewer parameters than the Spatial-Feature Factorized Linear Readout.

**Semantic Spatial Transformer Readout**  We introduce a novel readout that adaptively modulates both the encoder feature maps and their corresponding spatial weight distributions (i.e., receptive fields) on a per-voxel basis. Inspired by Spatial Transformer Networks (STN) (Jaderberg et al., 2015),

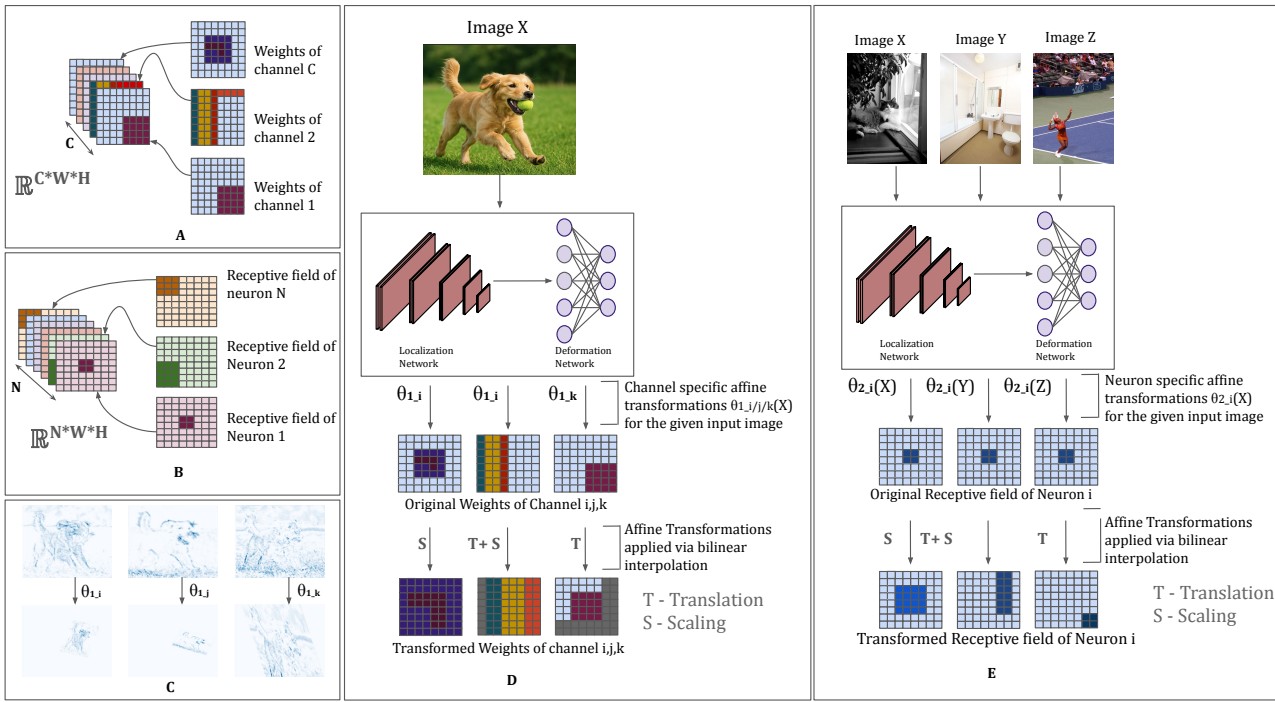

Figure 2: **Semantic Spatial Transformer Readout** (A) Schematic of encoder feature maps; color intensity reflects weight magnitude for interpretability. (B) Schematic of the learned Spatial Weights ("Where") matrix, which defines the receptive fields of the $N$ modeled neurons. Each neuron's receptive field is shown in a distinct color, with darker intensities highlighting its spatial extent (shape and location). (C–D) Input-dependent modulation of feature maps. (C) Example affine transformations applied to feature maps in response to the input image shown in (D). Top row: original feature maps; bottom row: corresponding transformed maps after applying learned affine transformations. Affine spatial transformations serve to reformat the features into a standardized canonical form on the fly, making the downstream processing more robust to variations such as scale, rotation, or translation. (D) Illustration of the pipeline for channel-specific spatial modulations: input $X$ induces different affine transformations across channels—e.g., channel $i$ undergoes scaling, channel $j$ experiences scaling and translation, and channel $k$ undergoes translation. (E) Input-dependent modulation of spatial receptive fields. The receptive field of the same neuron i is dynamically modulated based on different input stimuli X, Y, and Z. In these examples, the receptive field undergoes a scaling transformation for input X, a combination of scaling and translation for input Y, and translation for input Z.

this method spatially modulates the feature maps and spatial masks using affine transformations (e.g., rotation, scaling, and translation), allowing for dynamic and stimulus-dependent adjustments. The STN comprises two kinds of spatial modulations:

*Spatial modulation of spatial masks (Receptive Fields).* Unlike fixed spatial masks used in standard factorized or Gaussian readouts, our STN-based readout accommodates the dynamic nature of receptive fields (RFs). Biological evidence shows that RF sizes can expand or contract based on contrast (Sceniak et al., 1999) and can also shift or reshape in response to contextual or attentional cues (Womelsdorf et al., 2006). By allowing each voxel to learn its own affine transform, our method can capture such stimulus-dependent changes, moving beyond the static RF assumptions of conventional readouts (Figure 2 A, B, C).

*Spatial modulation of feature maps.* Beyond voxel-level RF

modulation, STNs also enable channel-wise transformations of the encoder features. Each feature channel may encode distinct visual attributes (e.g., edges, textures, or shapes) and thus might require unique spatial modifications. In contrast to object classification tasks—where known invariances (e.g., rotation, reflection) can be applied through data augmentation—voxel responses exhibit unknown geometric invariances. Allowing the network to learn channel-specific transforms directly from fMRI data provides a powerful mechanism to discover these invariances, potentially leading to richer and more accurate neural response models (Figure 2 D, E).

**STN Architecture -** Our STN module has four key components: **1. Localization Network -** A pretrained ResNet-50 that processes the raw stimulus image and outputs a feature representation before adaptive average pooling, **2. Linear Deformation Networks -** Two linear networks produce affine transformation parameters. From the localization features,

one generates $\theta_1 \in \mathbb{R}^{C \times 6}$ for the $C$ feature channels, while the other yields $\theta_2 \in \mathbb{R}^{N \times 6}$ for the $N$ voxels. Each row in $\theta_1$ and $\theta_2$ encodes a $2 \times 3$ matrix (6 parameters) for a unique affine transform, **3. Parameterized Sampling Grid -** Constructs sampling grids based on $\theta_1$ and $\theta_2$, defining how $\mathbf{E}$ (encoder feature map) and $\mathbf{S}$ (spatial weight matrix) are warped and **4. Sampler -** Applies bilinear interpolation to generate the transformed feature map $\mathbf{E}'$ and spatial weights $\mathbf{S}'$.

We compute each voxel's predicted response, $\hat{\mathbf{Y}}_n$, using the Spatial-Feature Factorized Linear Readout (Eq. 1), but replace $\mathbf{E}$ and $\mathbf{S}$ with their STN-transformed versions:

$$\mathbf{E}' = \mathrm{AT}(\mathbf{E}, \theta_1), \quad \mathbf{S}' = \mathrm{AT}(\mathbf{S}, \theta_2),$$

where $\mathrm{AT}(\mathbf{X}, \theta)$ applies a distinct $2 \times 3$ affine matrix in $\theta_m$ to each channel $m$ in $\mathbf{X} \in \mathbb{R}^{M \times W \times H}$. By jointly modulating receptive fields and feature channels, the STN readout captures the dynamic, context-dependent properties of neural responses and learns unknown geometric invariances directly from the data, offering a biologically motivated enhancement over fixed-mask readout methods. Further analysis on this readout is expanded in Appendix Table A4, Figure A6 and Section Analyzing spatial modulation of Receptive Fields in visual cortex: Insights from STN Readouts, where we examine how stimulus-dependent spatial shifts learned by the STN vary across the visual hierarchy. See Appendix Section Further Clarification on the pipeline for Semantic Transformers, Figures A7 and Table A8 for details on the individual affine transformations, computational complexity, and usability across different input stimuli.

### Training and Dataset

In this study, we utilized stimuli-response pairs from four subjects (Subjects 1, 2, 5, and 7) from the Natural Scenes Dataset (More details in Appendix section Natural Scenes Dataset). The experimental setup involved presenting a total of 37,000 image stimuli from the MS COCO dataset (Lin et al., 2014) to these subjects. Out of these, 1,000 images were shown to all four subjects, and these shared images were designated as the test set for our analyses. The remaining 36,000 images were split into 35,000 for training and 1,000 for validation purposes. We trained separate models for each of the following brain regions: the high-level ventral, dorsal and lateral streams, V4, V3v, V3d, V2v, V2d, V1v, and V1d. This approach allowed us to tailor the models to the unique neural response patterns of each region, thereby providing a more precise understanding of how different parts of the visual cortex process information. Throughout the paper, the reported accuracy refers to the test-time performance, measured as the noise-normalized Pearson correlation between predicted and actual voxel responses (see Appendix section Natural Scenes Dataset for noise ceiling computation).

All response-optimized models were trained using an NVIDIA GeForce RTX 4090 and NVIDIA A40 GPU. We employed a batch size of 4 with gradient accumulation to achieve an effective batch size of 16, using a learning rate of 0.0001.

Training was performed using an equal-weighted combination of Mean Squared Error (MSE) and correlation loss between predicted and target voxel responses, with early stopping applied after 20 epochs without improvement in validation accuracy, measured by Pearson correlation.

## Results

### Performance comparison of readouts across vision and language models in the visual cortex

We first evaluated the performance of various readout mechanisms in predicting neural responses across different brain regions. Our results showed that the Semantic Spatial Transformer readouts consistently outperform Linear, 2D Gaussian, and Spatial-Feature Factorized Linear readouts across all regions of the visual cortex and for almost all encoder models (see Figure 3). Its key advantage lies in the ability to flexibly adjust spatial masks and feature maps on a stimulus-by-stimulus basis, shifting receptive fields, resizing them, or rotating feature maps to align with a canonical form—transformations that better capture the actual variability in visual processing and boost predictive performance. This trend of superior performance is especially evident in vision models (see Figure 3-A) and holds for other task-optimized encoders processing visual input (details in Appendix Tables A1 and A2). Figure 3-B further illustrates the brain voxels where each readout performs best, underscoring the dominant performance of the Semantic Spatial Transformer readout for vision models across the visual hierarchy.

While the Semantic Spatial Transformer achieves the overall highest accuracy across all regions for all models (Appendix Tables A1, A2, A3), its improvement is less pronounced with language embedding inputs (Figure 3-B). This disparity arises because the Semantic Spatial Transformer readout uses a pretrained ResNet50 encoder as the localization network to learn affine transformations that adjust both vision and language encoder feature spaces. Vision encoder features are generally larger per channel (e.g., 28×28) than language encoder features (e.g., 4×4). Consequently, the Semantic Spatial Transformer readout has a greater capacity to leverage the rich spatial information available in vision models. Larger spatial dimensions provide more granular information, allowing STNs to learn transformations that account for variations in position, scale, and orientation of features more accurately. Further analysis on this bias introduced by readouts can be found in Appendix section Dependency of Semantic Spatial Transformer Readout on Channel Size and Table A5.

Further, Spatial-Feature Factorized Linear Readouts outperform Linear Ridge Regression Readouts both in terms of memory efficiency and prediction performance, as shown in Figure 3-A and Appendix Tables A3, A2 and A1. This improvement is attributed to the readout's capability to effectively disentangle voxel response selectivity into spatial and feature dimensions. This approach aligns with established phenomena in neuroscience, where neurons exhibit selectivity not only for specific features but also for stimuli presented within their re-

ceptive field locations.

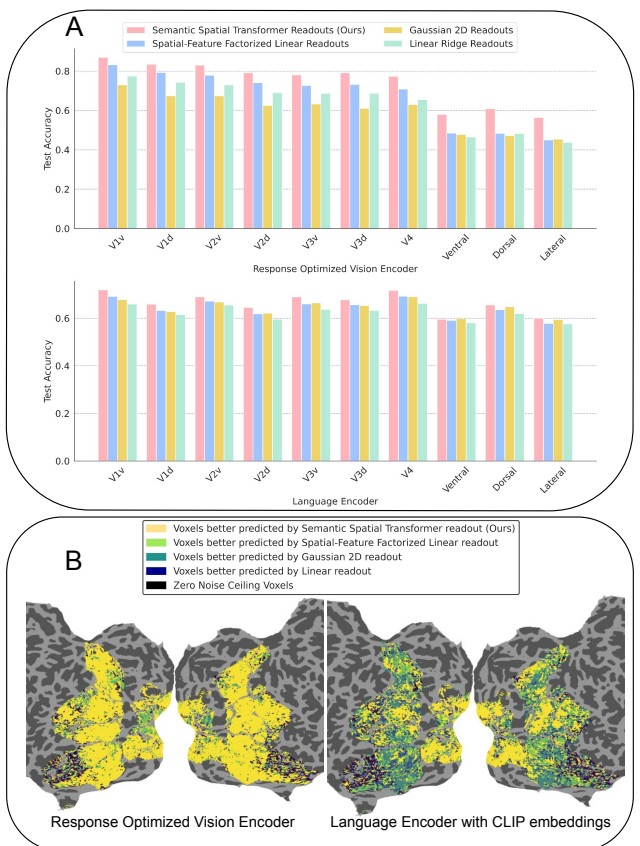

Figure 3: Comparison of readout mechanisms - (A) Noise Normalized Test Accuracy (Pearson Correlation) on held out dataset for different brain regions calculated using Response-optimized vision and Dense Language (CLIP embedding) models using four different readouts , (B) Brain visualizations showing regions where each readout performs the best

Gaussian 2D readouts are mostly outperformed by both Spatial-Feature Factorized Linear Readouts and Linear Ridge Regression Readouts in vision models, despite needing significantly fewer parameters. This performance gap can be attributed to the fact that Gaussian readouts were initially developed for grayscale stimuli in the mouse primary visual cortex (Lurz et al., 2020), where they utilized the brain's retinotopic mapping and anatomical organization to accurately define receptive fields. In our study, however, we learn the parameters of the Gaussian readout solely from the responses to complex image inputs, deliberately excluding anatomical information to maintain a fair comparison with other methods. Furthermore, this modeling approach may be less effective for the human visual system, where the assumption of a Gaussian-like structure may not hold true for the spatial receptive fields of all voxels, which may exhibit greater complexity.

Interestingly, the performance gap between Gaussian readouts and other readouts narrows in language models, where Gaussian readouts slightly outperform linear readouts across

all regions and exceed Spatial-Feature Factorized Linear Readouts in higher regions. This may be due to the smaller feature space in language models compared to vision models (e.g., 4×4 vs. 28×28), which simplifies receptive field localization.

Since the Semantic Spatial Transformer readout consistently outperformed others, we focus on it when analyzing the encoders in detail in the following sections.

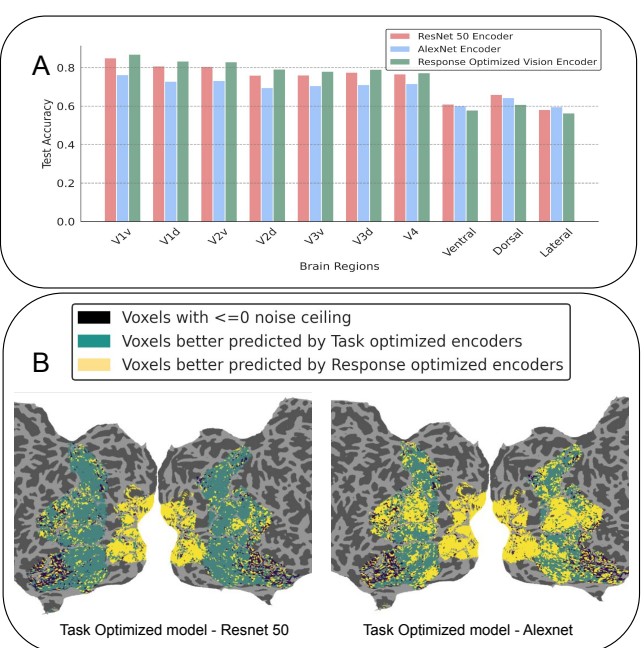

Figure 4: Comparison of Task-optimized versus Response-optimized vision models - (A) Test Accuracy (Normalized Pearson Correlation) on held out dataset using Task-optimized model encoders and Response-optimized model encoders with Semantic Spatial Transformer readout, (B) Brain visualization showing voxels better predicted by each model

## Task-optimized vs Response-optimized models

To ensure a fair comparison, we trained models using different sets of layers for each task-optimized model (Appendix Table A1 containing additional baselines ConvNext-Base Liu et al. (2022) and MOCO-V2 He et al. (2020)), and used only the best-performing ResNet50 layers for comparison, as presented in Table 1. In the early regions of the visual cortex (V1, V2, V3, and V4), response-optimized vision models consistently outperform task-optimized models by 2-12% (Figure 4 and Table 1), with a particularly notable margin over simpler architectures like AlexNet (Appendix Table A1). This suggests that features necessary for modeling early and mid-level visual areas are not fully captured by current task-optimized models, and explicit alignment with neural responsefs is crucial for higher prediction accuracy. This may be because task-optimized models, primarily trained on object-centric tasks, don't account for the broader range of visual

Table 1: Performance (Test Accuracies as Noise-Normalized Pearson Correlation) of Task-Optimized Vision models (ResNet-50, TV; best from A1), Response-Optimized Vision models (RV), and Language Models with CLIP embeddings—Single Caption (SL) and Dense Caption (DL). All use the Semantic Spatial Transformer readout, except SL which uses Ridge Linear readout.

| Model Details | V1v | V1d | V2v | V2d | V3v | V3d | V4 | Ventral | Dorsal | Lateral |
|---|---|---|---|---|---|---|---|---|---|---|
| TV | 0.8507 | 0.8083 | 0.8057 | 0.7603 | 0.7612 | 0.7763 | 0.7674 | **0.6105** | **0.6606** | 0.5823 |
| RV | **0.8698** | **0.8340** | **0.8302** | **0.7919** | **0.7808** | **0.7913** | **0.7729** | 0.5796 | 0.6089 | 0.5638 |
| SL | 0.3974 | 0.3779 | 0.3809 | 0.3702 | 0.4093 | 0.4119 | 0.4882 | 0.5661 | 0.6243 | 0.5920 |
| DL | 0.7196 | 0.6590 | 0.6903 | 0.6457 | 0.6897 | 0.6774 | 0.7167 | 0.5953 | 0.6562 | **0.6001** |

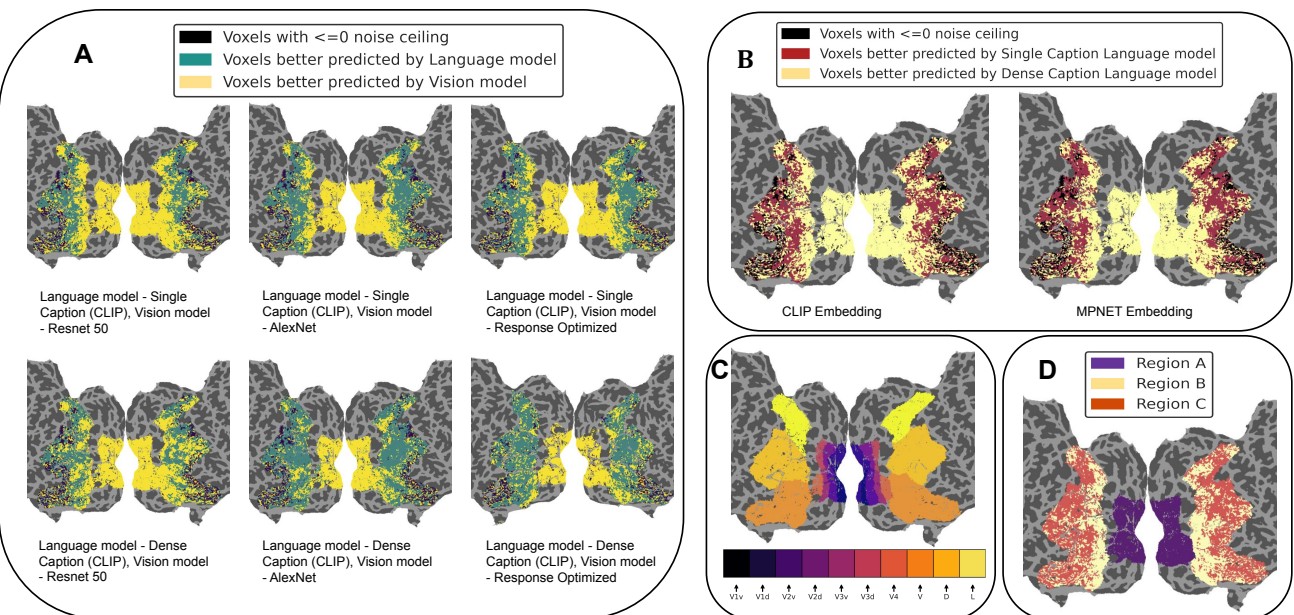

Figure 5: Comparison of vision and language models using Semantic Spatial Transformer readouts - Brain visualizations showing: (A) voxels better predicted by vision and language models, (B) voxels better predicted by single and dense caption language models, (C) the ten regions of the human visual cortex analysed in this study (V, D and L refer to Ventral, Dorsal and Lateral streams respectively), (D) highlighting three distinct regions, each demonstrating varying sensitivities to largely perceptual characteristics of the input, localized visual semantics aligned with linguistic descriptions, and global semantic interpretations of the input, also aligned with language

functions performed by the brain. Incorporating more ethologically relevant tasks into the optimization framework might be necessary for better modeling of early to mid-level visual processing. In the higher regions of the visual cortex (high-level ventral, dorsal, and lateral streams), task-optimized models show a slight performance advantage of around 5% over response-optimized models. This could be because these regions process more complex visual information, and task-optimized models, trained on larger object-centric datasets like ImageNet ($\geq$1.2 million images), better capture these functions. However, the small difference indicates that response-optimized models, despite being trained on only a fraction ( 3%) of the data, still capture significant aspects of high-level visual processing.

## Brain regions sensitive to vision vs language models

Recent research shows that pure language models, like MP-NET, can predict image-evoked brain activity in the high-level visual cortex using only image captions (Doerig et al., 2024). This raises intriguing questions about the alignment between the human visual cortex and language. To explore this relationship further, we compare these language with vision-only models.

When we assess language models that receive only image captions—without the images themselves—against response-optimized vision models, we find that the lower regions of the visual cortex are better modeled by vision-based approaches. In contrast, higher regions are more effectively captured by language models (see Figure 5-A, column 3 and Table 1). This pattern also holds when comparing language models to

task-optimized vision models, although the distinction is less pronounced (first two columns of Figure 5-A).

Next, we differentiate between single-caption and dense-caption models. Single-caption models convey only the overall semantic content of an image, whereas dense-caption models capture both spatial and semantic details. Consequently, the lower regions of the visual cortex, which are sensitive to fine-grained visual information, are better modeled by dense-caption models, as illustrated in Figure 5-B.

As we move from the lower to the higher regions of the visual cortex, there is a notable shift in sensitivity from localized semantics to global semantics across all ventral, dorsal, and lateral streams. Figure 5-B demonstrates that single-caption models dominate in the mid-to-higher regions of these streams, emphasizing the sensitivity of these areas to the overall meaning or interpretation of an entire image or scene. This trend is further corroborated in Figure 5-A, which compares vision models with both single-caption and dense-caption language models. Here, response-optimized vision models outperform single-caption models in the lower regions of the ventral, dorsal, and lateral streams, but do not maintain this advantage in the mid-to-higher regions.

Thus, we can identify three distinct regions in the visual cortex that are sensitive to different stimulus types (Figure 5-D): (1) lower visual regions (V1, V2, V3, and V4) are most sensitive to perceptual features that are not fully captured by linguistic descriptions - region A; (2) mid-level regions of the dorsal, ventral, and lateral streams are most sensitive to localized semantics (i.e. detailed, specific information about particular parts or regions of an image) - region B; and (3) higher regions of the dorsal, ventral, and lateral streams are sensitive exclusively to global semantic information - region C. Vision models outperform both single and dense caption language models in region A (Figure 5-A and Table 1), thus proving its sensitivity to largely perceptual features. Dense Caption language models outperform single caption language models (Figure 5-B) and response-optimized vision models (Figure 5-A) in region B, thus proving it is most sensitive to nuanced, localized semantic details. Vision models also outperform single caption models in region B (Figure 5-A), thus proving it is more sensitive to detailed visual information. Lastly, single caption language models outperform both dense caption models (Figure 5-B) and vision models (Figure 5-A) in region C, thus confirming its sensitivity to global semantics. Although this comparison was done mainly using Semantic Spatial Transformer readout, the trends hold true for other readouts, although to a much lesser extent (Appendix Figures A3, A2, A1).

## Discussion

In this study, we leveraged the NSD Dataset to evaluate various neural network models in predicting neural responses across different brain regions. Our analysis focused on three key comparisons: task vs. response optimized models, vision models vs. language models, and different readout methods for mapping model activations to brain signals.

First, we compared task-optimized models pre-trained on visual tasks (thus biased toward those tasks), with response-optimized models trained directly from brain response data. Response-optimized models significantly outperform task-optimized models in early visual regions. This suggests that brain-like processing in early-to-mid visual areas does not fully emerge in task-optimized models, and explicit alignment with neural data enhances prediction accuracy. However, in higher visual regions, both model types perform comparably, with task-optimized models showing a slight edge.

Next, we compared vision models with language models (both single-caption and dense-caption). Vision models outperformed language models in early visual regions, which are more attuned to perceptual features not captured by linguistic descriptions. In mid-level visual regions, sensitivity shifts toward semantic information, with dense-caption models excelling due to their ability to represent localized semantics. In higher visual regions, single-caption models perform better, indicating the importance of global scene understanding.

Finally, we evaluated different readout mechanisms for mapping activations to brain responses. Factorized readouts significantly outperformed standard linear models, and incorporating a Semantic Spatial Transformer further improved performance, particularly in vision models.

Our work has several limitations. First, we focused on task-optimized models trained for object categorization. A comprehensive comparison of models trained on other visual objectives and data sets is outside the scope of this study. However, prior research suggests that variations in architecture, objective, and data diet do not drastically impact response prediction accuracy (Conwell, Prince, Kay, et al., 2022), so we do not expect our conclusions to change significantly with additional models. While we found that language models become more accurate in predicting responses in high-level visual regions, we did not explore what specifically drives this performance Shoham et al. (2024), Conwell et al. (2023), Huh et al. (2024). It is still uncertain whether object category information (e.g., nouns) or other elements such as actions, spatial relationships, or contextual details play a more significant role. Finally, while the Semantic Spatial Transformer led to better predictions, future work should investigate how spatial and feature weights are modulated by different inputs. We also only tested affine transformations; more constrained or nonlinear deformations may offer further improvements.

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

# Appendix

## Natural Scenes Dataset

A detailed description of the Natural Scenes Dataset (NSD; http://naturalscenesdataset.org) is provided elsewhere (Allen et al., 2022). The NSD dataset contains measurements of fMRI responses from 8 participants who each viewed 9,000–10,000 distinct color natural scenes (22,000–30,000 trials) over the course of 30–40 scan sessions. Scanning was conducted at 7T using whole-brain gradient-echo EPI at 1.8-mm resolution and 1.6-s repetition time. Images were taken from the Microsoft Common Objects in Context (COCO) database (Lin et al., 2014), square cropped, and presented at a size of 8.4° x 8.4°. A special set of 1,000 images were shared across subjects; the remaining images were mutually exclusive across subjects. Images were presented for 3 s with 1-s gaps in between images. Subjects fixated centrally and performed a long-term continuous recognition task on the images. The fMRI data were pre-processed by performing one temporal interpolation (to correct for slice time differences) and one spatial interpolation (to correct for head motion). A general linear model was then used to estimate single-trial beta weights. Cortical surface reconstructions were generated using FreeSurfer, and both volume- and surface-based versions of the beta weights were created. In this study, we analyze manually defined regions of interest (ROIs) across both early and higher-level visual cortical areas. For early visual areas, we focus on ROIs delineated based on the results of the population receptive field (pRF) experiment - V1v, V1d, V2v, V2d, V3v, V3d, and hV4. For higher level visual cortex regions, we target the ventral, dorsal, and lateral streams, as defined by the streams atlas.

**Noise Ceiling Estimation in NSD** - Noise ceiling for every voxel represents the performance of the "true" model underlying the generation of the responses (the best achievable accuracy) given the noise in the fMRI measurements. They were computed using the standard procedure followed in (Allen et al., 2022) by considering the variability in voxel responses across repeat scans. The dataset contains 3 different responses to each stimulus image for every voxel. In the estimation framework, the variance of the responses, $\sigma^2_{\text{response}}$, are split into two components, the measurement noise $\sigma^2_{\text{noise}}$ and the variability between images of the noise free responses $\sigma^2_{\text{signal}}$.

$$\hat{\sigma}^2_{\text{response}} = \hat{\sigma}^2_{\text{signal}} + \hat{\sigma}^2_{\text{noise}}$$

An estimate of the variability of the noise is given as $\hat{\sigma}^2_{\text{noise}} = \frac{1}{n}\sum_{i=1}^{n} \text{Var}(\beta_i)$, where i denotes the image (among $n$ images) and $\text{Var}(\beta_i)$ denotes the variance of the response across repetitions of the same image. An estimate of the variability of the noise free signal is then given as,

$$\hat{\sigma}^2_{\text{signal}} = \hat{\sigma}^2_{\text{response}} - \hat{\sigma}^2_{\text{noise}}$$

Since the measured responses were z-scored, $\hat{\sigma}^2_{\text{response}} = 1$ and $\hat{\sigma}^2_{\text{signal}} = 1 - \hat{\sigma}^2_{\text{noise}}$. The noise ceiling (n.c.) expressed

Table A1: Performance (Test Accuracies as Normalized Pearson Correlation) of various Task Optimized vision models with Linear Ridge (R), Spatial-Feature Factorized Linear (F), Semantic Spatial Transformer (S) and Gaussian2D (G) readouts

| Model Details | | Visual Cortex Region | | | | | | | | | |
|---|---|---|---|---|---|---|---|---|---|---|---|
| Layers | Readout | V1v | V1d | V2v | V2d | V3v | V3d | V4 | Ventral | Dorsal | Lateral |
| colspan ResNet 50 |
| 1 | R | 0.6009 | 0.5695 | 0.5168 | 0.4783 | 0.4612 | 0.4543 | 0.4085 | 0.2958 | 0.3101 | 0.2508 |
| | G | 0.5935 | 0.5634 | 0.5110 | 0.4238 | 0.4135 | 0.4148 | 0.3758 | 0.2236 | 0.1928 | 0.1940 |
| | F | 0.8041 | 0.7627 | 0.7321 | 0.6950 | 0.6517 | 0.6540 | 0.5771 | 0.3252 | 0.3318 | 0.2709 |
| | S (Ours) | 0.8498 | 0.8022 | 0.7860 | 0.7501 | 0.7559 | 0.7461 | 0.7410 | 0.5763 | 0.6208 | 0.5652 |
| 2 | R | 0.5618 | 0.6535 | 0.6276 | 0.4677 | 0.4564 | 0.4485 | 0.4157 | 0.3628 | 0.3796 | 0.3131 |
| | G | 0.6478 | 0.5827 | 0.5694 | 0.5086 | 0.4975 | 0.4861 | 0.4898 | 0.2958 | 0.2797 | 0.2593 |
| | F | 0.8142 | 0.7728 | 0.7601 | 0.7302 | 0.6956 | 0.7110 | 0.6403 | 0.4034 | 0.4116 | 0.3515 |
| | S (Ours) | **0.8507** | **0.8083** | **0.8057** | **0.7603** | **0.7612** | **0.7763** | 0.7601 | 0.5813 | 0.6241 | 0.5667 |
| 3 | R | 0.6599 | 0.6413 | 0.6426 | 0.6014 | 0.6051 | 0.6237 | 0.6138 | 0.5022 | 0.5657 | 0.4689 |
| | G | 0.6607 | 0.6110 | 0.6359 | 0.5920 | 0.6270 | 0.6205 | 0.6526 | 0.4991 | 0.5296 | 0.4671 |
| | F | 0.8046 | 0.7666 | 0.7705 | 0.7482 | 0.7465 | 0.7675 | 0.7540 | 0.5751 | 0.6277 | 0.5384 |
| | S (Ours) | 0.7898 | 0.7393 | 0.7643 | 0.7193 | 0.7496 | 0.7495 | **0.7674** | **0.6105** | **0.6606** | **0.5823** |
| 4 (all) | R | 0.2812 | 0.2577 | 0.2583 | 0.2556 | 0.2880 | 0.2433 | 0.3132 | 0.3006 | 0.2922 | 0.2820 |
| | G | 0.5170 | 0.4671 | 0.4810 | 0.4318 | 0.4821 | 0.4787 | 0.5442 | 0.4764 | 0.4702 | 0.4704 |
| | F | 0.5922 | 0.5488 | 0.5606 | 0.5297 | 0.5542 | 0.5659 | 0.5612 | 0.4525 | 0.4741 | 0.4269 |
| | S (Ours) | 0.6989 | 0.6504 | 0.6746 | 0.6487 | 0.6743 | 0.6791 | 0.6814 | 0.5857 | 0.6337 | 0.5809 |
| colspan AlexNet |
| 1 | R | 0.6359 | 0.6320 | 0.5844 | 0.5403 | 0.5268 | 0.5178 | 0.4795 | 0.3134 | 0.3275 | 0.2727 |
| | G | 0.6520 | 0.6009 | 0.5539 | 0.5197 | 0.4649 | 0.4489 | 0.4550 | 0.3156 | 0.3054 | 0.2662 |
| | F | 0.7253 | 0.6897 | 0.6479 | 0.6136 | 0.5678 | 0.5841 | 0.5300 | 0.3170 | 0.3178 | 0.2763 |
| | S (Ours) | 0.7590 | 0.7159 | 0.7229 | 0.6662 | 0.6934 | 0.6764 | 0.7004 | 0.5594 | 0.6072 | 0.5556 |
| 2 | R | 0.5822 | 0.5550 | 0.5268 | 0.4951 | 0.4919 | 0.4855 | 0.4715 | 0.2924 | 0.2949 | 0.2485 |
| | G | 0.6459 | 0.6221 | 0.5883 | 0.5489 | 0.5439 | 0.5357 | 0.5278 | 0.3688 | 0.3399 | 0.3271 |
| | F | 0.7325 | 0.6923 | 0.6704 | 0.6396 | 0.6168 | 0.6287 | 0.5876 | 0.3864 | 0.3822 | 0.3322 |
| | S (Ours) | 0.7710 | **0.7288** | 0.7273 | 0.6950 | 0.7043 | **0.7117** | **0.7169** | 0.5705 | 0.6002 | 0.5408 |
| 3 | R | 0.5951 | 0.5722 | 0.5554 | 0.5260 | 0.5234 | 0.5313 | 0.5197 | 0.3389 | 0.3392 | 0.2879 |
| | G | 0.6311 | 0.6150 | 0.5997 | 0.5705 | 0.5597 | 0.5629 | 0.5719 | 0.4289 | 0.4072 | 0.3888 |
| | F | 0.7419 | 0.7055 | 0.7033 | 0.6713 | 0.6655 | 0.6864 | 0.6594 | 0.4618 | 0.4711 | 0.4098 |
| | S (Ours) | **0.7634** | 0.7236 | **0.7327** | **0.6961** | **0.7065** | 0.7092 | 0.7148 | 0.5694 | 0.6071 | 0.5326 |
| 4 | R | 0.6145 | 0.5830 | 0.5834 | 0.5527 | 0.5733 | 0.5677 | 0.5632 | 0.4123 | 0.4181 | 0.3486 |
| | G | 0.6357 | 0.6038 | 0.5994 | 0.5677 | 0.5684 | 0.5795 | 0.5908 | 0.4601 | 0.4827 | 0.4220 |
| | F | 0.7325 | 0.6933 | 0.6989 | 0.6724 | 0.6735 | 0.6895 | 0.6758 | 0.5066 | 0.5323 | 0.4559 |
| | S (Ours) | 0.7444 | 0.7070 | 0.7173 | 0.6816 | 0.7037 | 0.7108 | 0.7129 | 0.5688 | 0.6214 | 0.5458 |
| 5 (all) | R | 0.4931 | 0.4798 | 0.4703 | 0.4474 | 0.4560 | 0.4609 | 0.4662 | 0.3717 | 0.3806 | 0.3394 |
| | G | 0.5605 | 0.5652 | 0.5136 | 0.5193 | 0.4946 | 0.5231 | 0.5260 | 0.4523 | 0.4334 | 0.4201 |
| | F | 0.6889 | 0.6339 | 0.6679 | 0.6183 | 0.6602 | 0.6502 | 0.6833 | 0.5803 | 0.6347 | 0.5797 |
| | S (Ours) | 0.7168 | 0.6653 | 0.6859 | 0.6481 | 0.6855 | 0.6797 | 0.7156 | **0.6003** | **0.6443** | **0.5965** |
| colspan ConvNext Base |
| 1 | S | **0.8238** | 0.7744 | 0.775 | 0.7324 | 0.7334 | 0.7361 | 0.7257 | 0.5579 | 0.5948 | 0.5186 |
| 2 | S | 0.8194 | **0.7762** | **0.7753** | **0.7425** | **0.7520** | **0.7595** | **0.7536** | 0.5745 | 0.6172 | 0.5521 |
| 3 | S | 0.6697 | 0.6345 | 0.6445 | 0.6057 | 0.6504 | 0.6600 | 0.6858 | **0.5756** | 0.6392 | 0.5836 |
| 4 (all) | S | 0.6688 | 0.6209 | 0.6394 | 0.5914 | 0.6394 | 0.6351 | 0.6742 | 0.5761 | **0.6374** | **0.5679** |
| colspan Moco V2 |
| 1 | S | 0.8379 | 0.7735 | 0.7898 | 0.7370 | 0.7505 | 0.7459 | 0.7552 | 0.5828 | 0.6083 | 0.5635 |
| 2 | S | **0.8405** | **0.7967** | **0.8018** | **0.7630** | **0.7632** | **0.7736** | 0.7589 | 0.5917 | 0.6366 | 0.5758 |
| 3 | S | 0.8066 | 0.7604 | 0.7791 | 0.7340 | 0.7589 | 0.7722 | **0.7649** | 0.6082 | **0.6621** | **0.5850** |
| 4 (all) | S | 0.7111 | 0.6586 | 0.6793 | 0.6457 | 0.6735 | 0.6790 | 0.6980 | **0.6038** | 0.6641 | 0.6019 |

Table A2: Performance (Test Accuracies as Normalized Pearson Correlation) of Response Optimized vision models with Linear Ridge (R), Spatial-Feature Factorized Linear (F), Semantic Spatial Transformer (S) and Gaussian2D (G) readouts

| Readout | V1v | V1d | V2v | V2d | V3v | V3d | V4 | Ventral | Dorsal | Lateral |
|---|---|---|---|---|---|---|---|---|---|---|
| R | 0.7746 | 0.7427 | 0.7299 | 0.6906 | 0.6867 | 0.6865 | 0.6551 | 0.4657 | 0.4824 | 0.4372 |
| G | 0.7306 | 0.6744 | 0.6746 | 0.6253 | 0.6326 | 0.6104 | 0.6297 | 0.4784 | 0.4728 | 0.4545 |
| F | 0.83154 | 0.7926 | 0.7795 | 0.7419 | 0.7268 | 0.7323 | 0.7085 | 0.4847 | 0.4831 | 0.4504 |
| S (Ours) | **0.8698** | **0.8340** | **0.8302** | **0.7919** | **0.7808** | **0.7913** | **0.7729** | **0.5796** | **0.6089** | **0.5638** |

Table A3: Performance (Test Accuracies as Normalized Pearson Correlation) of language models (C: CLIP, M: MPNET, G-XL: GPT2-XL) with Linear Ridge (R), Spatial-Feature Factorized Linear (F), Semantic Spatial Transformer (S) and Gaussian2D (G) readouts

| Model Details | | Visual Cortex Region | | | | | | | | | |
|---|---|---|---|---|---|---|---|---|---|---|---|
| LLM | Readout | V1v | V1d | V2v | V2d | V3v | V3d | V4 | Ventral | Dorsal | Lateral |
| Single Caption Models | | | | | | | | | | | |
| C | R | **0.3974** | **0.3779** | **0.3809** | **0.3702** | **0.4093** | **0.4119** | **0.4882** | 0.5661 | 0.6243 | 0.5920 |
| M | R | 0.3931 | 0.3738 | 0.3738 | 0.3687 | 0.4031 | 0.4077 | 0.4873 | **0.5672** | **0.6269** | **0.6126** |
| G-XL | R | 0.3791 | 0.3642 | 0.3653 | 0.3540 | 0.3953 | 0.4036 | 0.4773 | 0.5638 | 0.6162 | 0.6007 |
| Dense Caption Models | | | | | | | | | | | |
| C | R | 0.6597 | 0.6154 | 0.6551 | 0.5953 | 0.6371 | 0.6322 | 0.6621 | 0.5807 | 0.6201 | 0.5761 |
| | G | 0.6783 | 0.6277 | 0.6682 | 0.6207 | 0.6644 | 0.6531 | 0.6905 | 0.5980 | 0.6491 | 0.5943 |
| | F | 0.6919 | 0.6329 | 0.6721 | 0.6183 | 0.6603 | 0.6572 | 0.6927 | 0.5915 | 0.6365 | 0.5781 |
| | S (Ours) | 0.7196 | 0.6590 | 0.6903 | 0.6457 | 0.6897 | 0.6774 | 0.7167 | 0.5953 | **0.6562** | 0.6001 |
| M | R | 0.6557 | 0.5941 | 0.6325 | 0.5732 | 0.6162 | 0.6207 | 0.6493 | 0.5679 | 0.5831 | 0.5502 |
| | G | 0.6840 | 0.6261 | 0.6659 | 0.6207 | 0.6583 | 0.6519 | 0.6928 | 0.5934 | 0.6441 | 0.5894 |
| | F | 0.6889 | 0.6339 | 0.6679 | 0.6183 | 0.6602 | 0.6502 | 0.6833 | 0.5803 | 0.6347 | 0.5797 |
| | S (Ours) | 0.7168 | 0.6653 | 0.6859 | 0.6481 | 0.6855 | 0.6797 | 0.7156 | **0.6003** | 0.6443 | 0.5965 |
| G-XL | R | 0.6738 | 0.6272 | 0.6586 | 0.6136 | 0.6625 | 0.6504 | 0.6862 | 0.5881 | 0.6380 | 0.5732 |
| | G | 0.6895 | 0.6284 | 0.6717 | 0.6203 | 0.6605 | 0.6631 | 0.6980 | 0.5941 | 0.6501 | 0.6003 |
| | F | 0.6940 | 0.6386 | 0.6716 | 0.6275 | 0.6597 | 0.6636 | 0.6974 | 0.5874 | 0.6381 | 0.5832 |
| | S (Ours) | **0.7253** | **0.6653** | **0.7038** | **0.6619** | **0.6956** | **0.6939** | **0.7242** | 0.5974 | 0.6487 | **0.6023** |

in correlation units is thus given as $n.c. = \sqrt{\frac{\hat{\sigma}^2_{signal}}{\hat{\sigma}^2_{signal}+\hat{\sigma}^2_{noise}}}$. The models were evaluated in terms of their ability to explain the average response across 3 trials (i.e., repetitions) of the stimulus. To account for this trial averaging, the noise ceiling is expressed as $n.c. = \sqrt{\frac{\hat{\sigma}^2_{signal}}{\hat{\sigma}^2_{signal}+\hat{\sigma}^2_{noise}/3}}$. We computed noise ceiling using this formulation for every voxel in each subject and expressed the noise-normalized prediction accuracy (R) as a fraction of this noise ceiling.

## Unimodal versus multimodal embeddings in language models

As outlined in the previous section, the higher-level regions of the ventral, dorsal, and lateral visual streams exhibit heightened sensitivity to broad semantic information that captures the overall meaning of a scene, as opposed to specific visual details or a combination of visual and spatial features. These regions are best modeled by single-caption language models. To investigate this further, we examine the performance of models using unimodal encoders like MPNET, which are trained exclusively on language, and multimodal encoders like CLIP, trained on both language and visual data. In the higher regions of the ventral, dorsal, and lateral streams, models using MPNET encoders slightly outperform those with CLIP encoders by 0.5%. This marginal advantage in the higher regions may be attributed to MPNET's optimization for capturing rich semantic nuances from text, aligning well with the language-sensitive nature of these brain regions. On the other hand, in the lower visual regions, where responses are more strongly driven by visual inputs, CLIP encoders hold a small advantage of 1% over MPNET, likely due to their integration of visual knowledge. However, this trend does not hold in dense caption language models, where the performance of both encoders is comparable.

## The Necessity of Spatial Subdivision in Dense Captioning for Effective Visual Cortex Modeling

We further investigated whether the observed differences between dense and global captioning are due to (a) the spatial

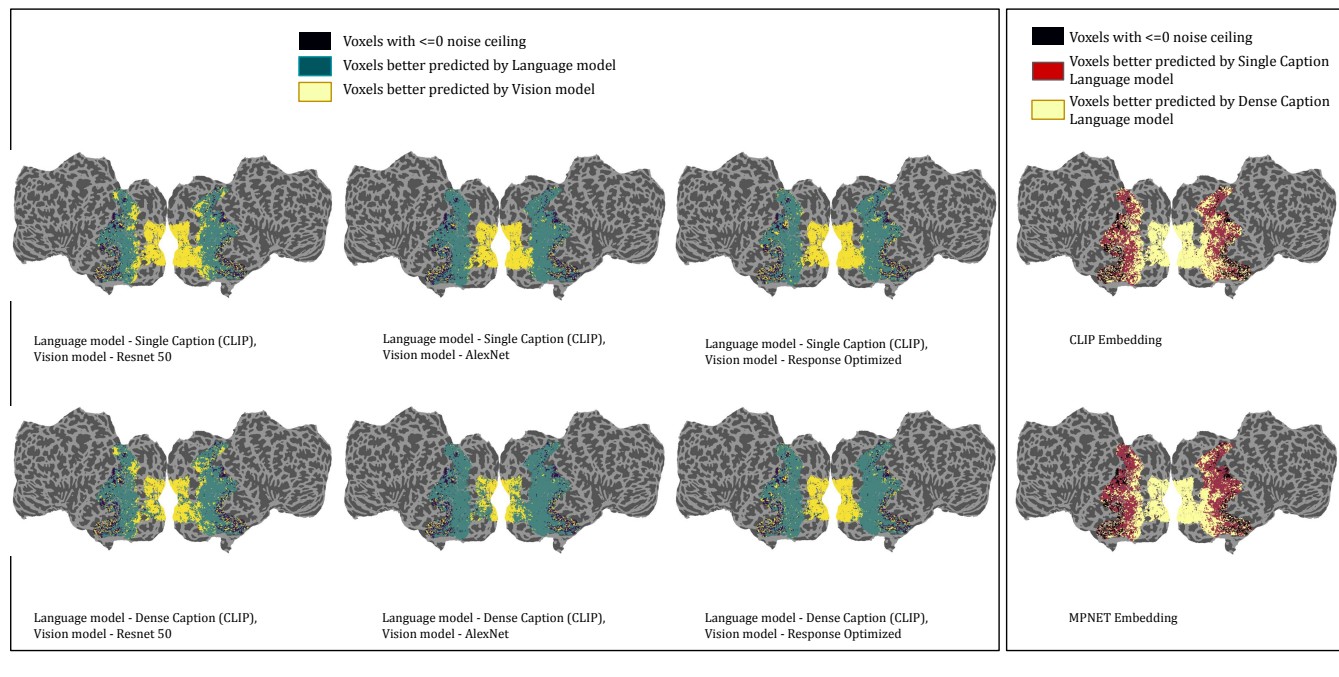

**A**

**B**

Figure A1: A - Brain Visualizations showing voxels that are better predicted by vision and language models, all using Ridge Linear readouts, B - Brain Visualizations showing voxels that are better predicted by single caption and dense caption language models, all using Ridge Linear readouts

subdivision of the image (Hypothesis 1) or the increased semantic detail in dense captions (Hypothesis 2). The original idea behind using dense captions was to provide spatial information in addition to semantic information in the form of captions, and subdividing the image into equal sized grids and getting captions for each grid was one of the easiest and most intuitive ways to do that.

We further tried generating more comprehensive single captions of the image using existing LLMs, however none of them were able to provide more information than those already present in the original MS-COCO dataset. In an attempt to densify the single captions, we thus adopted a different approach: for each image, we took the embeddings of dense captions generated for individual grid locations and averaged these embeddings to produce a single "aggregate dense caption" embedding.

On comparing single caption stimuli with 'densified' single caption stimuli (as opposed to the dense caption approach discussed in the paper) (Figure A5), we saw a similar trend where the higher regions of the visual cortex were better modeled by single caption stimuli. However, the transition in sensitivity from dense to single caption in the middle regions of the ventral, dorsal and lateral stream that is so clearly pronounced when using dense captions is missing when using the above 'densified' single captions. Further comparing 'densified' single captions to dense captions (as proposed in the paper), we

saw that the dense captions modeled the overall visual cortex better. Hence, we do feel that adding spatial information to the dense caption is necessary for building more accurate models, be it by sub-dividing the image into grids or via any other way.

## Analyzing spatial modulation of Receptive Fields in visual cortex: Insights from STN Readouts

In an additional experiment focused on interpreting the STN readouts, we calculated the distance between the affine parameters corresponding to the spatial maps of each voxel for every image, relative to the mean affine parameters across all images (Figure A6). The L2 norm of this vector was computed for each voxel. Across all encoders, we observed that stimulus-dependent spatial shifts of the receptive field increase from lower to higher visual regions. A similar trend emerged when calculating the average spatial shifts for each channel of the feature map across images for different regions. This trend further supports the idea that higher levels of the visual cortex benefit more from learned geometric invariances and exhibit greater spatial modulation of their visual receptive fields compared to lower visual cortex regions. This modulation includes phenomena such as receptive field expansion, contraction, or shifts in response to different stimuli.

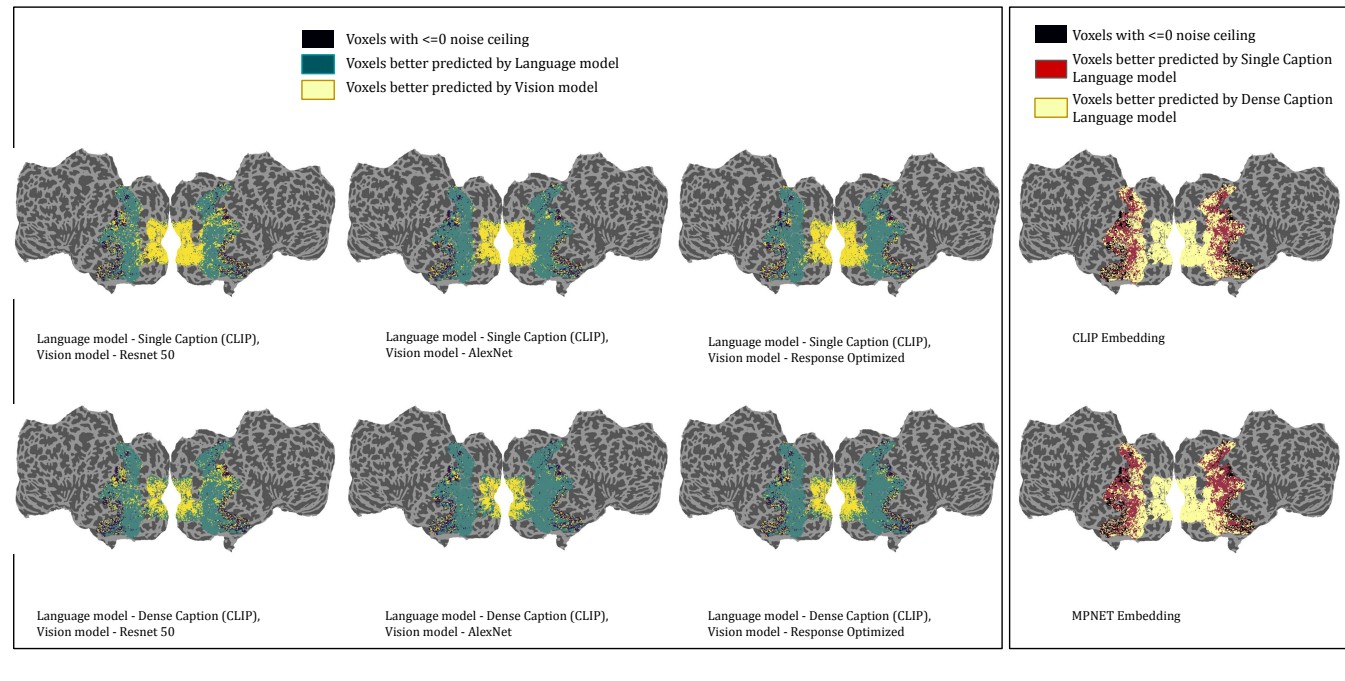

**A**

**B**

Figure A2: A - Brain Visualizations showing voxels that are better predicted by vision and language models, all using Gaussian2D readouts, B - Brain Visualizations showing voxels that are better predicted by single caption and dense caption language models, all using gaussian2D readouts

Table A4: Performance (Test Accuracies as Normalized Pearson Correlation) of Spatial-Feature Factorized Linear Readout (F) with individual affine transformations applied to encoder feature maps (1) and spatial masks (2) separately, all with Response Optimized Vision models.

| Readout | V1v | V1d | V2v | V2d | V3v | V3d | V4 | Ventral | Dorsal | Lateral |
|---|---|---|---|---|---|---|---|---|---|---|
| F | 0.83154 | 0.7926 | 0.7795 | 0.7419 | 0.7268 | 0.7323 | 0.7085 | 0.4847 | 0.4831 | 0.4504 |
| F + 1 | 0.8596 | 0.8217 | 0.8179 | 0.7769 | 0.7705 | 0.7719 | 0.7659 | 0.5638 | 0.5962 | 0.5371 |
| F + 2 | 0.8750 | 0.8409 | 0.8310 | 0.7948 | 0.7814 | 0.7958 | 0.7782 | 0.5865 | 0.6156 | 0.5641 |

Table A5: Performance (Analysis of the effect of channel size on the improvement introduced by Semantic Spatial Transformer Readout (S) over Spatial-Linear Factorized Readouts (F), all with Response Optimized Vision models

| Readout | V1v | V1d | V2v | V2d | V3v | V3d | V4 | Ventral | Dorsal | Lateral |
|---|---|---|---|---|---|---|---|---|---|---|
| F (28*28) | 0.8315 | 0.7926 | 0.7795 | 0.7419 | 0.7268 | 0.7323 | 0.7085 | 0.4847 | 0.4831 | 0.4504 |
| S (28*28) | 0.8698 | 0.8340 | 0.8302 | 0.7919 | 0.7808 | 0.7913 | 0.7729 | 0.5796 | 0.6089 | 0.5638 |
| S (4*4) | 0.8432 | 0.8089 | 0.8056 | 0.7690 | 0.7672 | 0.7743 | 0.7425 | 0.5734 | 0.5986 | 0.5513 |
| S (4*4) | 0.7783 | 0.7328 | 0.7374 | 0.6991 | 0.7061 | 0.7043 | 0.7102 | 0.5699 | 0.6002 | 0.5532 |

## Dependency of Semantic Spatial Transformer Readout on Channel Size

We acknowledge the importance of ensuring that the readout does not skew conclusions about neural representations. The larger improvements for vision models stem from their feature representations having greater spatial dimensions than lan-guage models, allowing the SST to better leverage the rich spatial information available in vision models. To mitigate this, we can normalize spatial dimensions across models to ensure uniform treatment. Empirically we show that if we reduce the spatial dimensions of the vision encoder to match those of the language encoder, that does drop the prediction performance

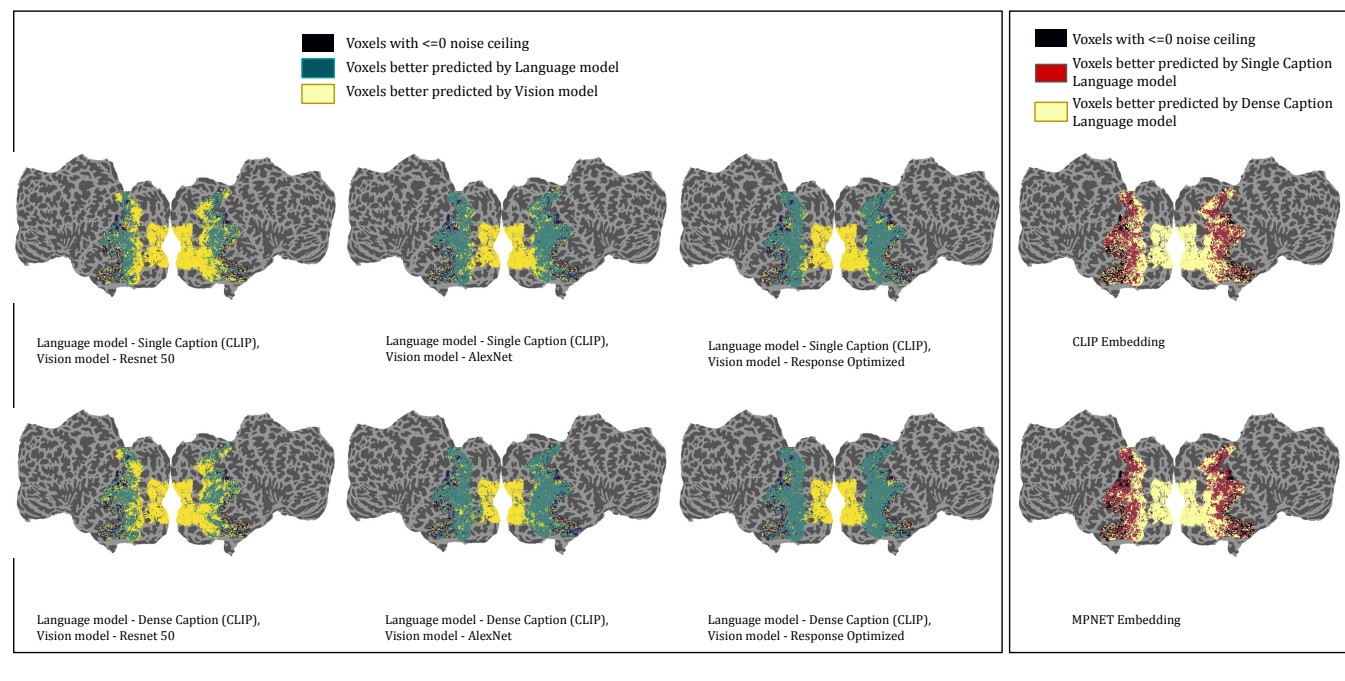

**A**

**B**

Figure A3: A - Brain Visualizations showing voxels that are better predicted by Vision and language models, all using Spatial-Feature Factorized Linear readouts, B - Brain Visualizations showing voxels that are better predicted by single caption and dense caption language models, all using Spatial-Feature Factorized Linear readouts

and relative gains (Table A5).

The overall trend where higher cortical areas are better modeled by language input and lower cortical areas by visual input is consistently observed across all readouts (Figure. 5, A1, A2, A3). However, the margin distinguishing the effectiveness of the models varies slightly. Notably, as we progress from less biologically intuitive readouts to more biologically plausible ones (linear regression, Gaussian 2D, Spatial-Feature Factorized Linear Readout, and finally, the Semantic Spatial Transformer Readout), these trends become increasingly well-defined. Given that the Semantic Spatial Transformer Readout most accurately and consistently models neural responses, we rely on it to delineate regions of the visual cortex sensitive to varying kinds of stimulus information.

### Comparing different architectures for Task and Response Optimized models

Our study carefully controlled several factors to compare task-optimized and response-optimized neural network models for predicting brain responses. Specifically, we held constant both the stimulus set and readout layer, varying only the encoder architecture across models. The rationale for employing different architectures in our study was to leverage state-of-the-art approaches tailored to distinct modeling paradigms. A direct comparison between task-optimized and response-optimized models is inherently challenging due to differences

in the available training stimulus sets. Specifically, the stimulus set for training response-optimized models is substantially smaller—approximately 0.03 times the size of the datasets used for task optimization (e.g. ImageNet). Incorporating structural biases into response-optimized models (e.g., rotation equivariance) enables them to learn effectively from smaller datasets. This advantage of rotation-equivariant architectures in neural encoding contexts has been demonstrated in prior studies Khosla & Wehbe (2022) and is a critical factor when designing models that align with the constraints of neural data.

While head-on comparisons using identical architectures for task and neural response optimization could provide valuable insights into the specific contributions of these factors , the primary objective of our study was not to isolate these factors. Instead, we aimed to identify the most predictive models for voxel responses across distinct regions of the visual system. Our findings reveal the current best-performing models for this goal, emphasizing practical predictive utility rather than dissecting the contributions of task versus response optimization in isolation.

We conducted further experiments using - a ResNet-50 encoder trained from scratch exclusively on the NSD dataset, a Mask-RCNN encoder trained from scratch on the NSD dataset, a pretrained Mask-RCNN encoder finetuned on the NSD dataset, and compared it with the proposed task and

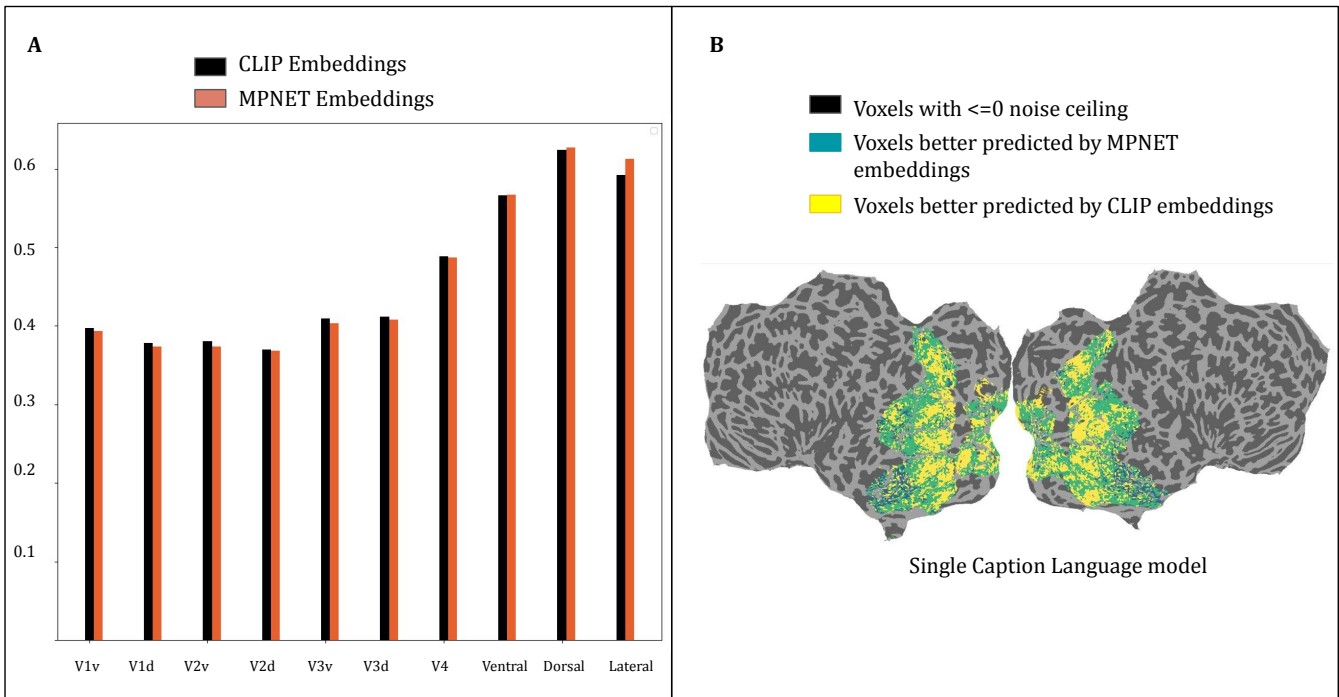

Figure A4: Comparison of Unimodal and Multimodal embeddings in Language models, A - Test Accuracy (Normalized Pearson Correlation) on held out dataset using Single Caption Language encoders with CLIP and MPNET embeddings, B - Brain Visualization showing regions better predicted by each encoder in Single Caption Language models

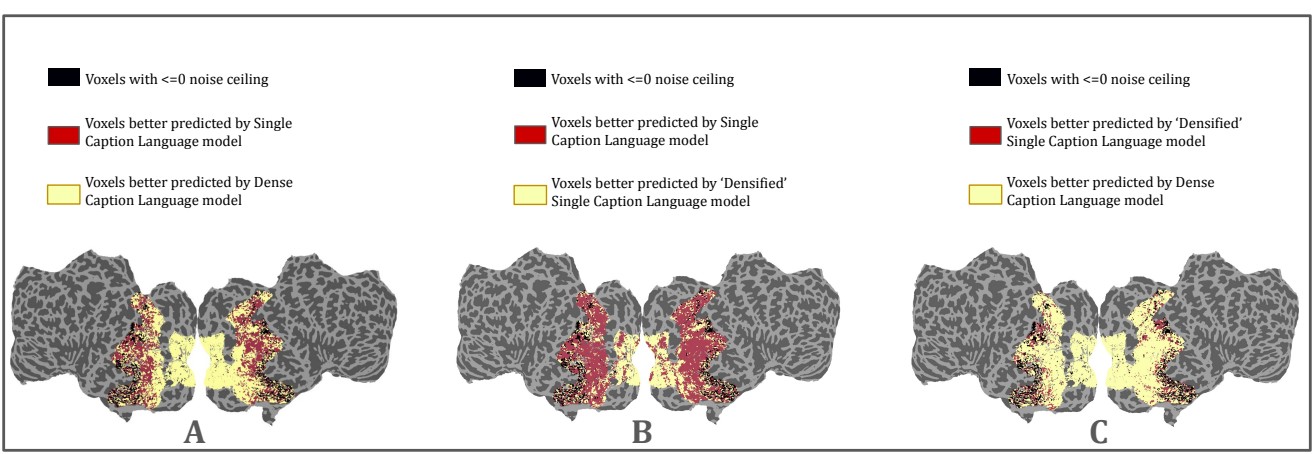

Figure A5: A - Comparison of Single Caption Language models with Dense Caption Language models, B - Comparison of Single Caption Language models with 'Densified' Single Caption Language model, C - Comparison of 'Densified' Single Caption Language model with Dense Caption Language model

response optimized encoders in the paper all paired with a Semantic Spatial Transformer readout (Table A6). We did this to analyze if the same architecture for response- and task-optimized vision models could provide valuable insights. Unlike the task-optimized ResNet-50, which is trained for ob-

ject classification on ImageNet, the ResNet-50 trained from scratch on neural responses struggled to match the performance of the proposed response-optimized e2cnn model. The task optimized Mask-RCNN model is pretrained on the MS-COCO dataset which is a superset of the images in the

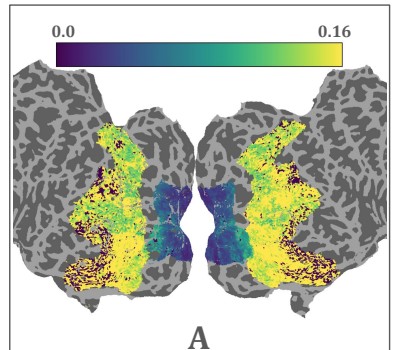
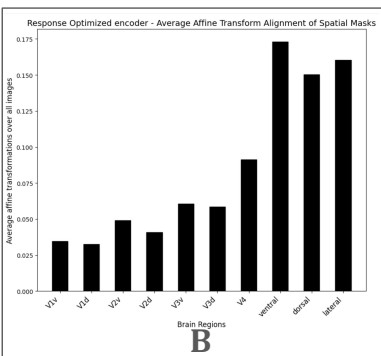
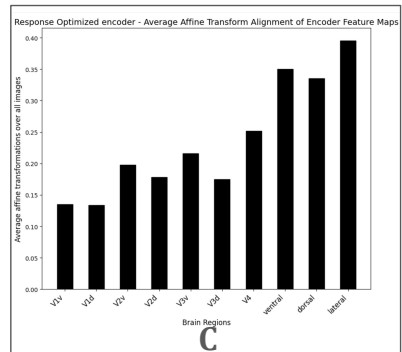

Figure A6: A - Average spatial shifts of voxel spatial masks across all images, B - Mean spatial shifts for each brain region, comparing spatial masks across all images, C - Mean spatial shifts for each brain region, comparing feature maps across all images.

Table A6: Performance (Analysis of different architectures for Response and Task Optimized models (A - Task Optimized Resnet 50 (pretrained with ImageNet), B - Response Optimized Resnet 50, C - Task Optimized Mask-RCNN (pretrained with MS-COCO), D - Response Optimized Mask-RCNN, E - Response Optimized E2cnn (proposed), all with Semantic-Spatial Transformer Readouts.

| Encoder Type | V1v | V1d | V2v | V2d | V3v | V3d | V4 | Ventral | Dorsal | Lateral |
|---|---|---|---|---|---|---|---|---|---|---|
| A | 0.8507 | 0.8083 | 0.8057 | 0.7603 | 0.7612 | 0.7763 | 0.7674 | 0.6105 | 0.6606 | 0.5823 |
| B | 0.7579 | 0.7034 | 0.7021 | 0.6646 | 0.6861 | 0.6712 | 0.6991 | 0.5546 | 0.5814 | 0.5470 |
| C | 0.8543 | 0.8144 | 0.8084 | 0.7693 | 0.7680 | 0.7772 | 0.7793 | 0.6077 | 0.6764 | 0.5987 |
| D | 0.8147 | 0.7654 | 0.7621 | 0.7163 | 0.7089 | 0.6898 | 0.7114 | 0.5648 | 0.5841 | 0.5469 |
| E | 0.8698 | 0.8340 | 0.8302 | 0.7919 | 0.7808 | 0.7913 | 0.7729 | 0.5796 | 0.6089 | 0.5638 |

NSD dataset. Although both the task optimized performance show a very similar performance, we once again see a similar trend here with the Mask-RCNN encoder trained from scratch on the NSD dataset, where it struggled to reach the performance of the response optimized e2cnn model. This comparison underscores the role of network architecture and the significance of incorporating relevant structural biases into networks when optimizing them on response prediction with limited data (atleast in comparison to large-scale vision datasets).

Task-optimized models, typically pretrained on large-scale datasets (e.g., ImageNet), apply only a linear mapping from their learned representations to brain responses. Although one could examine how diverse architectures and tasks affect performance, prior work (Conwell, Prince, Alvarez, & Konkle, 2022; Conwell et al., 2024) suggests that even starkly different architectures (e.g., CNNs vs. transformers) yield similar brain predictivity in task-optimized settings, implying that architecture alone may not be the critical factor. Here, we take a complementary approach by comparing these task-optimized models with response-optimized and LLM-based frameworks,

each configured to best align with neural data constraints. Specifically, we select the most effective pretrained architecture for task optimization and pair it with an appropriately chosen architecture for response optimization.

**Further Clarification on the pipeline for Semantic Transformers**

Figure A7 presents an overview of the pipeline when using the Semantic Spatial Transformer Readout. This readout builds upon the existing Spatial-Feature Factorized readout, whose components are highlighted in the orange box in the figure. The key innovation introduced by the Semantic Spatial Transformer is the application of affine transformations to both the encoder feature representation and the spatial weight ("where") matrix, enabling data augmentation and modulation of receptive fields dynamically based on the input. To enable these transformations, the readout incorporates four additional components: (1) Localization Network (2) Deformation Network (seperate for each affine transformation set) (3) Parameterized Sampling Grid (4) Sampler.

The localization network is implemented using a pretrained

Table A7: Performance (Test Accuracies as Normalized Pearson Correlation) of Single Caption Language models - (1) entire sentence is used, (2) Only object words are used, (3) Only stuff words are used, (4) Both object and stuff words are used and (5) Jumbled sentences are used

| Caption Type | V1v | V1d | V2v | V2d | V3v | V3d | V4 | Ventral | Dorsal | Lateral |
|---|---|---|---|---|---|---|---|---|---|---|
| 1 | 0.3974 | **0.3779** | **0.3809** | **0.3702** | **0.4093** | **0.4119** | **0.4882** | **0.5661** | **0.6243** | 0.5920 |
| 2 | 0.3342 | 0.3252 | 0.3197 | 0.322 | 0.3407 | 0.3506 | 0.4051 | 0.4905 | 0.5287 | 0.5187 |
| 3 | 0.3186 | 0.2603 | 0.2888 | 0.2530 | 0.2973 | 0.2868 | 0.3413 | 0.4237 | 0.4418 | 0.4143 |
| 4 | 0.3721 | 0.3316 | 0.3439 | 0.3242 | 0.3609 | 0.3575 | 0.4200 | 0.5038 | 0.5415 | 0.5178 |
| 5 | **0.4016** | 0.3759 | 0.3802 | 0.3682 | 0.4080 | 0.4099 | 0.4840 | 0.5615 | 0.6211 | **0.5952** |

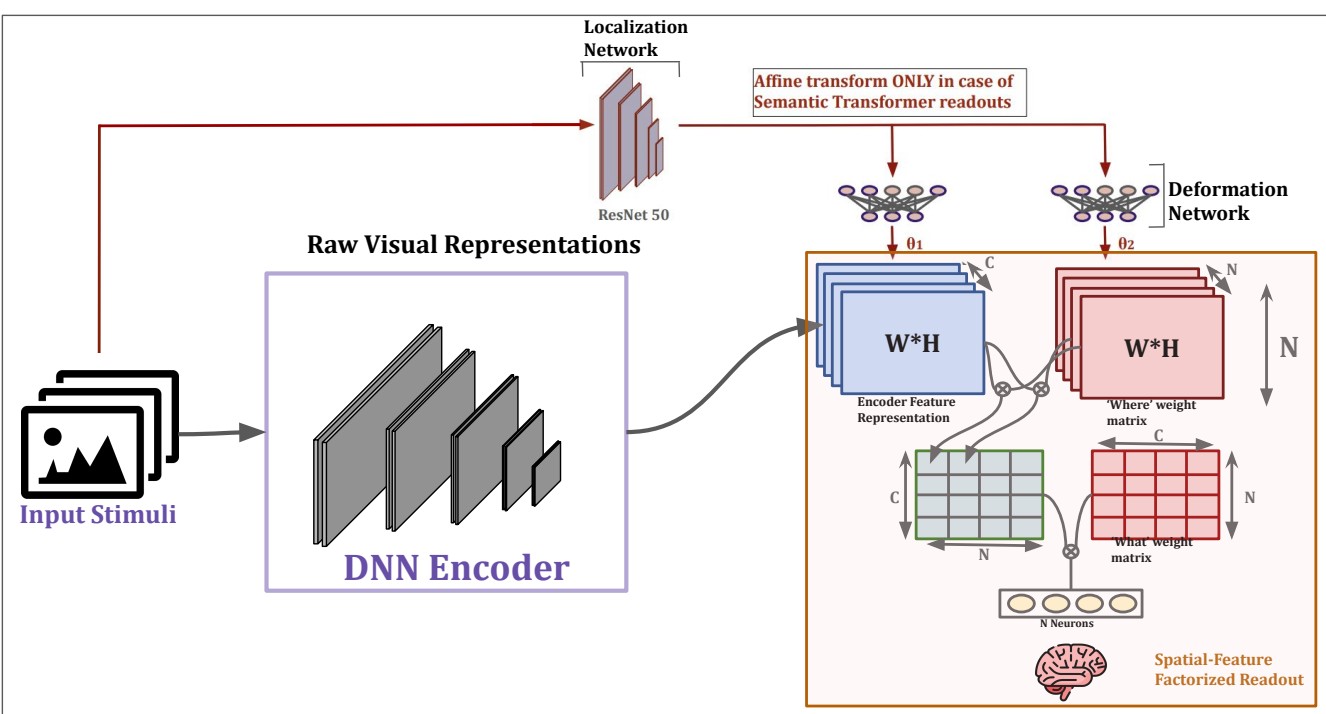

Figure A7: Overall Pipeline when a Semantic Spatial Transformer readout is used.

Table A8: Number of learnable parameters for each readout configuration - Here, $C$ denotes the number of channels in the encoder feature representation, $N$ is the number of neurons being modeled, and $WH$ represents the spatial dimensions of each feature map channel.

| Readout Type | Number of parameters learnt |
|---|---|
| Linear | $N * C * H * W$ |
| Gaussian 2D | $N * (C + 7)$ |
| Spatial-Feature Factorized | $N * (C + W * H)$ |
| Semantic Transformer | $N * (C + W * H) + 32 * 6(N + C) + 196 * 32 * 2$ |

ResNet-50 block, which generates input stimulus embeddings. Importantly, this network's weights are frozen during training. The motivation for using a pretrained network is to leverage strong, prior-informed embeddings, which can facilitate the learning of effective affine transformations. While the main DNN encoder in the pipeline (whether task-optimized or response-optimized) could also serve as a localization network, we chose a fixed pretrained model to ensure robust and stable representations. Incorporating the main encoder as the localization network is a promising direction for future work.

Each of the two deformation networks is implemented as a linear layer that receives embeddings from the localization

network and outputs 6-parameter affine transformations for two distinct purposes -

1. $\theta_1$: transformation parameters for each channel of the encoder feature representation ($R^{C*W*H}$), to apply stimuli dependent data augmentations on each channel.

2. $\theta_2$: transformation parameters for each neuron in the spatial weight ("where") matrix ($R^{N*W*H}$), that will modulate the respective neuron's receptive field based on the input stimuli.

Once $\theta_1$ and $\theta_2$ are obtained, they are applied to the respective $W \times H$ grids using PyTorch's built-in affine-grid (to generate sampling grids) and grid-sample (to apply the transformations) functions. The parameterized sampling grid defines how each location in the transformed grid corresponds to coordinates in the original grid. For example, a target coordinate $(x, y)$ in the transformed space might map back to a source coordinate $(i, j)$ in the original grid. Since these source coordinates may not align perfectly with discrete pixel locations, the Sampler uses bilinear interpolation to compute the output value at $(x, y)$ by interpolating values from neighboring pixels around $(i, j)$ in the input.

The affine transformations applied to the encoder feature representations (parameterized by $\theta_1$) for data augmentation purposes are further illustrated in Figure 2 A,C,D. Similarly, the transformations applied to the spatial weight matrix (parameterized by $\theta_2$), which allow for dynamic modulation of receptive fields, are detailed in Figure 2 B, E.

**Computational complexity of the Semantic Spatial Transformer Readout** The Semantic Spatial Transformer introduces minimal overhead—the extra complexity comes solely from two lightweight deformation networks that predict affine transformation parameters for each feature channel in the encoder and one for each voxel. The localization network is configured to output embeddings of dimension 196. Each deformation network starts with a linear layer that projects this 196-dimensional embedding to a hidden dimension of 32, which is then further transformed into a 6-parameter affine transformation. The total number of learnable parameters in each deformation network is -

1. For $\theta_1$ (channel-wise transformations): $196*32+32*6*C$, where C is thte total number of channels.

2. For $\theta_2$ (neuron-wise transformations): $196*32+32*6*N$, where N is the number of neurons.

The additional parameters (roughly $32 \cdot 6 \cdot (N+C)$ plus a constant term) are modest relative to the overall parameter count of the encoder. Moreover, the affine grid generation and bilinear sampling operations are computationally efficient and scale linearly with the feature map size. Table A8 summarizes the number of parameters that need to be learned for each readout configuration.

**How are dense caption stimuli used with Semantic Transformer readouts?** To generate dense caption stimuli, the original image (e.g., of size $424 \times 424$) is first divided into uniform patches of size $8 \times 8$, resulting in a grid of $53 \times 53$ chunks. For each chunk, a caption is generated using a language model (e.g., GPT-2). These captions are then embedded into vector representations using a large language model (LLM). Let the embedding dimension be $M$, which varies depending on the LLM used—for example, $M = 512$ for CLIP, $M = 768$ for MPNET, and $M = 1600$ for GPT-2 XL. As a result, the dense caption stimuli can be interpreted as an "image" of shape $M \times 53 \times 53$, analogous to a standard RGB image of shape $3 \times 424 \times 424$.

Dense caption stimuli are specifically used in conjunction with a 2-block E2CNN encoder, similar to the response-optimized models used for visual stimuli (which typically use 8 blocks). The output of this encoder is a set of feature maps that can be represented as a $C \times W \times H$ matrix, which integrates naturally with the "what" and "where" matrices in the Semantic Spatial Transformer Readout. To generate the affine transformations, we do not pass the dense caption stimuli directly. Instead, the original image stimuli are passed through the ResNet-50 localization network to produce more robust and semantically meaningful affine parameters. This design choice is motivated by the desire to leverage strong visual priors from pretrained models. A promising direction for future work would be to investigate whether affine transformations can be learned directly from linguistic descriptions alone, without relying on the original visual input.

