# OpenReview forum: "Modeling the Human Visual System: Comparative Insights from Response-Optimized and Task-Optimized Vision Models, Language Models, and different Readout Mechanisms"
_ccneuro.org/CCN/2025/Proceedings — CCN 2025 Proceedings asProceedingsPoster_

### Official Review · Reviewer_UdYa · 2025-03-28
**A well-written paper with novel methodological contributions, but limited model comparisons**

**Soundness:** 2
**Clarity:** 3

**Comments:**

This paper aims to compare both a wide variety of NN encoding models and readout mechanisms in their ability to match visual cortex response. The authors introduce a new readout mechanism, which outperforms other methods such as linear readout in most voxels in visual cortex. In addition, they find that response-optimized models are more predictive of low-level visual regions while task-optimized and LLM models are more predictive in more anteorior regions.

The paper is overall clearly written and makes interesting methodological contributions. However, I have some concerns about the novelty and support for the scientific claims, as well as some technical clarifications. See specific comments in rough order of importance below.

1. Relatively few encoder models in each class are tested. For example the only task-optimized vision models reported are Alexnet and Resnet, both of which are relatively old / small models. Would similar trends be observed with larger models (including contrastively trained models)? This limitation makes it difficult to draw conclusions about response-optimized versus task-optimized models, particularly given that the models likely vary along several factors (architecture, training size, etc.)

2. A similar issue when comparing models across modality. A given vision and language model vary across many dimensions, and either largescale or controlled comparisons are needed to make a strong inference about a given model factor (see Conwell 2024 Nat Comms; Conwell 2023 CCN; Wang 2023 Nat Machine Intelligence, for discussion).

3. While the comparison between response and task optimized models is novel, the conclusion that engineering optimized models are better at matching high-level visual responses is unsurprising. Engineering optimized vision models have been the dominant method for modeling high-level visual cortex responses since Yamins 2014.

4. The STN readout seems highly novel and perhaps the most interesting aspect of the paper. However, while the individual model components are well motivated, the model seems extremely complicated, consisting of multiple separate trained neural networks. Why is this a fair or interesting comparison with the other readout mechanisms?

5. While the overall motivation for the STN was very clear, some of the technical aspects outlined in 343-357 were dense and difficult to follow. It would help to flesh these out and improve the illustrative figure in 1D to be more information (perhaps using an actual image as in 1B).

6. It was not entirely clear how the STN readout could be applied to language model encoders. This is discussed on 425-422 but this was difficult to follow.

7. The issue of neural readout has been discussed at two different CCN GACs. It might help to cite/refer to these (Ivanova et al., 2022 NBDT; Conwell et al 2024 CCN)

**Expertise:**

3

**Interest:**

3

---

> ### Author Rebuttal · Authors · 2025-04-15
>
> Thank you for the constructive feedback. Below, we address all seven comments in a coherent order rather than by number. Most suggested changes are added to the appendix of the revised version. Where appropriate, we will move some to the main manuscript in the camera-ready version, though time constraints prevent that for now.
>
> 5,6: Appendix: "Further Clarification on the Pipeline for Semantic Transformers" and new figures A7–A9 detail the implementation.
>
> 4: Simplicity: SST readout only adds stimulus-based affine transforms to existing feature and spatial maps (introduced previously), using lightweight 2 layer MLPs. These learn to modulate specific channels and voxels based on the input, without complex nonlinearities.
> Our readout is biologically motivated - it dynamically adjusts receptive fields based on the stimulus, consistent with known neural behavior (lines 320–330). Aligning feature maps into a canonical pose improves robustness to scale, rotation, and translation (lines 331–342). The strong performance gains with these simple affine transformations indicate they capture essential aspects of neural response modeling.
>
> 1,3: Prior work [1] showed diverse architectures and training paradigms yield similar brain predictivity for task-optimized (TO) models, motivating our concise baseline set. We added ConvNeXt Base and MoCo v2—differing in size, architecture, and objectives (Table A1). These results (with SST readout) reinforce our core finding: early to mid visual regions favor response-optimized (RO) models, while higher regions slightly favor TO ones.
> Although TO models align with high-level responses, our findings and [1] show architecture has limited impact on their predictivity. In contrast, RO models benefit from structural biases—e.g., rotation equivariance boosts performance—highlighting how tailored architectures offset limited data. (Appendix: “Comparing different architectures for Task and Response Optimized models”).
> [1]Conwell et al., 2021, [2] Yamins 2014
>
> 2: Precisely matching modalities is challenging. For vision, alongside our established baselines like ResNet and AlexNet, we added new baselines—ResNet-MoCo (Table A1), which uses a distinct learning objective, and ConvNeXt, representing a different architectural family—in response to the reviewer’s suggestion for additional baselines. Refer to response 1 to Reviewer eya5 for further details on additional experiments on language models.
>
> 7: We discuss them now in line 108 and 69.

---

> > ### Comment · Reviewer_UdYa · 2025-04-17
> >
> > Thank you for your detailed responses and edits. I believe the manuscript is significantly improved and I have adjusted my scores. In particular, the additional analyses on SST are very interesting and add to this contribution and the additional baseline models help strengthen the task versus data-optimization claims..
> >
> > I have two remaining suggestions:
> >
> > 1. The SST methods section could still be improved/clarified in the main text without adding length. eg - "Two linear networks" could be replaced with "two layer MLPs" as you write in the rebuttal.
> >
> > 2. I understand the modality comparison is difficult. Given these limitations (not just in this work but other similar model comparisons) the authors should add some discussion of these issues and/or consider tempering the claims about modality/"semantic" advantages.

---

> > > ### Author Response · Authors · 2025-04-18
> > >
> > > Thank you for your thoughtful feedback and helpful suggestions. However, we believe the updated review scores may not have been reflected yet—please pardon us if we're mistaken. If this was an oversight, we would kindly request the reviewer to update their score if possible.
> > >
> > > Regarding the camera-ready version, we will incorporate the following updates:
> > > - We will replace “Two linear networks” with “Two layer MLPs” in the final version for greater clarity.
> > >
> > > - As suggested, we will expand the discussion of limitations, specifically the paragraph spanning lines 608–627. This will involve revisions to meet the 8-page limit.
> > >
> > > - Additionally, we will include results for the Linear, Gaussian2d, and Spatial-Feature Factorized Linear readouts for the MoCo-V2 and ConvNeXt-Base baselines in Table A1, to further strengthen our analysis of task-optimized models.
> > >
> > > Once again, thank you for your constructive input.

---

### Official Review · Reviewer_eYa5 · 2025-03-29
**This paper presents a comparison of encoder-readout combinations for modeling the human visual cortex. Its main contributions include a novel Semantic Spatial Transformer Readout mechanism and the identification of three functionally distinct regions in the visual cortex that process visual information differently. The work provides valuable insights into how different brain regions process perceptual vs. semantic information, with significant results. However, limitations include reliance on a single dataset and limited exploration of what specific linguistic features drive model performance in higher visual regions. Overall, this represents an important exploration in computational modeling of visual processing**

**Soundness:** 3
**Clarity:** 2

**Comments:**

This manuscript presents a systematic comparison of encoders (task-optimized, response-optimized, and language models) and readout mechanisms for predicting visual cortex responses to natural scenes. The authors introduce a novel Semantic Spatial Transformer Readout (SSTR) that dynamically adjusts receptive fields based on image content. Their analysis identifies three functionally distinct regions in the visual cortex with different sensitivities: lower regions responsive to perceptual features, mid-level regions sensitive to localized semantics, and higher regions attuned to global semantic information.

**Strengths:**
- The SSTR approach significantly outperforms existing methods (3-23% over factorized methods, 7-53% over ridge regression)
- Comprehensive evaluation of multiple encoder-readout combinations across brain regions on NSD dataset
- Interesting insights into visual cortex organization based on sensitivity to perceptual vs. semantic information
- Important contributions to understanding language-vision integration in neural processing

**Limitations:**
- Reliance on a single dataset (NSD) limits generalizability claims
- Lack of deeper analysis into what specific linguistic features drive language models' performance
- Computational complexity of SSTR not thoroughly assessed

**Questions:**
1. How does SSTR's computational complexity compare to simpler readout methods?
2. How exactly are captions generated for each grid cell in the dense-caption approach?
3. How do your three functional regions relate to established category-selective regions?
4. Have you tested generalizability across other fMRI datasets?

**Expertise:**

2

**Interest:**

3

---

> ### Author Rebuttal · Authors · 2025-04-15
>
> Thank you for the feedback. We've addressed all comments, with most changes in the appendix of the revised version. Where appropriate, we will move some to the main manuscript in the camera-ready version, though time constraints prevent that for now.
>
> 1. Rather than focusing on specific linguistic features, we aimed to identify brain regions where either vision/language inputs dominate, revealing sensitivity to perceptual cues, localized semantics, or global meaning.   For language models, we compared MPNET ( masked language modeling objective enhanced by permutation-based predictions,) and CLIP (contrastive objective to align text with images) language encoder, and we now also include GPT2-XL (trained autoregressively to predict the next token, Table A3).  To probe which linguistic elements best model the visual cortex, we also tested sentence manipulations including content words and jumbled versions (Table A7). Notably, the original sentence consistently delivers the best or near-best performance, suggesting that full-sentence baselines provide a stronger foundation for visual cortex modeling than representations based solely on selected or reordered linguistic elements.
>
> 2. We provide a detailed explanation of the computational complexity of the Semantic Spatial Transformer readout (SSTR) in the Appendix section titled “Further Clarification on the Pipeline for Semantic Transformers” (Lines 1168–1190), including  a comparison with other readouts in Table A8.  In short, SSTR adds only a minimal overhead of $32\times6\times(N+C)$ parameters (C=# channels, N=#voxels)—a modest increase wrt the encoder's total parameters—along with efficient affine grid generation and bilinear sampling operations.
>
> 3. Captions are generated using GPT-2 for each grid in the dense caption stimuli(lines 233-236).
>
> 4. Category-selective regions (ex- FFA for faces) are localized within the ventral stream and are selective for specific object categories. In contrast, our three functional regions span the full visual hierarchy, from early to higher-order areas (containing category-selective ROIs), and are defined by their sensitivity to representational complexity rather than category selectivity.
>
> 5. Few publicly available fMRI datasets capture responses across the visual cortex to large-scale, naturalistic scenes. The Natural Scenes Dataset (NSD) remains the only dataset with both high spatial resolution and extensive trial counts, making it the standard in NeuroAI research.

---

> > ### Comment · Reviewer_eYa5 · 2025-04-21
> >
> > I appreciate the authors' responses and clarifications. The revised paper presents a valuable contribution to the community, and I now strongly support its acceptance.

---

> > > ### Author Response · Authors · 2025-04-21
> > >
> > > Thank you for the encouraging feedback and strong endorsement of our paper. It appears that the updated review scores may not have been reflected yet—please pardon us if we're mistaken. If this was an oversight, we would kindly request the reviewer to update their score if possible to match their revised assessment.

---

### Official Review · Reviewer_jbu7 · 2025-03-31
**Solid, methodologically rich, and compelling work; though could still perhaps benefit from greater focus on metrics other than performance...**

**Soundness:** 3
**Clarity:** 3

**Comments:**

**A note of disclosure**: I reviewed (at least, I'm pretty sure) an earlier version of this work submitted to a previous conference, and am (for the sake of time) copying / pasting some of the content there, but have attempted in due diligence to try and "diff" the two versions and focus on the improvements in this one! TLDR: I had voted for acceptance of the previous version, and feel even more confident in doing so with the version submitted here.

(Previous Summary): In this work, the authors present a comprehensive suite of analyses comparing vision / language DNN models to human fMRI data. Using a novel “readout” mechanism designed explicitly to account for space in the mapping of DNN embeddings to brain activity, the authors report localizing 3 sub-regions in the human visual cortex that respond differentially to spatial and semantic information.

(Previous) Overall Review: The use of deep neural network models to predict and understand the structure of representation in the biological visual system is a practice rife with heretofore unanswered, but deeply foundational questions as to how it should be done. Bucking a trend that far too often recycles canonical, but relatively unscrutinized methods to new models or new brain data, this submission is impressive not just for the fact that it tackles these questions head-on, but tackles so many of them simultaneously -- and does so (mostly) without losing the forest for the trees. For this alone, I applaud the authors and can recommend that this paper be accepted.

(Updated) Overall Review: Much the same, but now I recommend acceptance with even greater confidence!

Questions / comments from the previous review that I think this version inadequately / adequately / compellingly addresses:

**"Have we learned anything new about the visual system?"**
(Previous) Review: My major concern here (and one that I admit is not fully within the authors control, but which clarifying updates or different narrative focus could nonetheless address) is the lingering doubt as to whether even these newer, more expertly designed methods actually do give us any meaningful new “insights” about the biological system they’re nominally designed to give us insights about. An overly reductionist summary of the “findings” of this analysis with respect to the human visual brain could well be that they simply provide more evidence for what is already a amply established gradient of increasingly “abstract” visual information from early (more view-dependent) areas (where smaller, localized receptive fields and retinotopy are the dominant representational motifs) to later (less view-dependent) visual areas (where -- depending on which side of the ventral / dorsal divide those areas are closer to -- you begin to get “representations” that evoke “object categories”, “navigational affordances”, or “conceptual semantics”). And while much debate does remain as to many of the details here, it seems (to me at least) that the existence of this gradient is more or less a common consensus.)

(Updated) Review: I find this work compelling enough and the method so well-motivated that I am less keen to dwell on this, though I think my concern remains largely the same... With so carefully considered a modeling methodology, surely we must see something beyond differences in performance! Reading this improved version, I found myself wondering if perhaps I was intuitively looking for something like a double dissociation (Model A has X competence; Model B has Y competence; Model A can predict X phenomenon in Area 1, but not Y phenomenon in Area 2; Model B can predict Y phenomenon in Area 1, but not X phenomenon in Area 2, et cetera...)? Or perhaps (as the authors previously took care to note) a bit more detail about the kinds of representational information the "semantic spatial transformer" can capture that other models have previously not. I think the authors do indeed have this, but it's not as evident in the paper as it could be... (More on this below).

**Is there something here beyond improved performance?"**
(Previous Review): “Beyond accuracy”: The primary justification of the authors’ “novel readout mechanism” is the general increase in accuracy it provides over other methods. But the emphasis on accuracy as the primary advantage rings a bit hollow if a major part of the goal here is to gain insight into the structure of representation in biological cortex. There are many alternative ways (e.g. data augmentation, denoising, nonlinearities) -- even “hacky” ones (e.g. smaller cross-validation splits) that one could use to increase the predictive accuracy of model readout mechanisms. What demonstrable advantage does the “semantic spatial transformer” readout have over readout methods with respect to the theoretical questions at play here?...

(Updated Review): Accuracy remains the focus of this paper, and I do understand why (considering the "semantic spatial transformer" is very much a new method that some in the field might disregard if it weren't for its increased predictive accuracy), but it seems to me there might be so much more! Lifting (if I may be so bold) a response written by the authors to my previous review, I find this snippet particularly intriguing: "The STN also spatially transforms feature maps (i.e. the encoder channels). Each channel typically encodes distinct attributes (e.g., edges, textures, shapes), and the ability to apply channel-specific transformations allows the STN to adapt to their unique geometric properties. For instance, one channel may require scaling to emphasize fine-grained details, while another might need rotation for orientation invariance...." This is lovely! But it naturally leaves me asking questions such as: "which properties? which features? which channels?" It seems the "semantic spatial transformer" might actually be giving us substantive and theoretically interesting details about *how* the model representations are being transformed into increasingly accurate predictors of the voxel activity -- and by association, giving us information about the representations inherent to the voxel activity. I find myself dreaming of flatmaps with detailed tuning curves, feature visualizations, receptive field sizes, and all the other information the "semantic spatial transformer" seems to be picking up on more than other models. Should this be saved for follow-up work? Perhaps... Nevertheless, I leave the idea here as food for thought.

**"What is the language model predictivity *really* saying about visual system representation?"**
(Previous Review): “Semantics” without language model confounds: There are a number of issues (again, beyond the scope of this paper, but nonetheless relevant) with the use of language models as predictive models of visual fMRI data -- including the fact the inputs to these models (tokenized words) are already proto-symbolic at the time of their initial injection into the candidate representational models that embed them (and are thus more abstract by default than the pixels injected into vision models); and also, an increasing “convergence” between vision and language models [1] that suggests a sort of “default” alignment between these systems attributable (most likely, it seems) to biases in their training data....

(Updated Review): I appreciate the updates the authors have made to their discussion of language model predictivity, and will rest my case for now, though I encourage the authors to consider [2, 3] as additional sources of discussion.

**"What's going on with those dense captions?"**
(Previous Review): “Densifying” the “single” captions: The authors claim that the localized semantic descriptions inherent to their “dense” captioning method unveil a noticeable midpoint between early, more spatiotopic representations and later, “globally” abstract representations. But is this really about the local tagging of an image’s subparts? Providing more comprehensive “single captions” of the full image that includes more extensive specification of details might close the gap between the dense captioning method and the global captioning -- but in a way that obviates the need to manually subdivide the image. In short, adding further detail (with and without explicitly spatial language) seems like an important control for downstream interpretation of this result.

(Updated Review): I applaud the authors on an additional experiment + supplementary section the authors have added since my last review -- assuming I'm looking at the right version, yikes! -- that I think very amply addresses my interest in / concern with this set of analyses. There is some follow-up work to be done, perhaps, with NLP manipulations of the captions to further hone in on what aspects of the "dense" captions are most directly contributing to the improved performance over the "densified" single captions the authors have now compared against, but that can wait!

**Final Thoughts**: CCN would benefit greatly from seeing this paper presented! **Strong accept!**

[1] Huh, M., Cheung, B., Wang, T., & Isola, P. (2024). The platonic representation hypothesis. arXiv preprint arXiv:2405.07987.

[2] The Unreasonable Effectiveness of Word Models in Predicting Visual Cortex Responses to Natural Images (CCN 2023)

[3] Shoham, A., Broday-Dvir, R., Malach, R., & Yovel, G. (2024). The organization of high-level visual cortex is aligned with visual rather than abstract linguistic information. bioRxiv, 2024-11.

**Expertise:**

2

**Interest:**

3

---

> ### Author Rebuttal · Authors · 2025-04-15
>
> Thank you for your kind and encouraging feedback. It’s challenging to encapsulate everything within the scope of a single conference paper, but many of the directions you've highlighted are indeed part of our ongoing and future work.
>
>
> In particular, we aim to further investigate what exactly is learned by our proposed Semantic Spatial Transformer Readout—specifically, how it modulates feature maps and receptive fields in response to different stimuli, and which features are emphasized in this process. For instance, does the nature of this modulation vary across different brain regions? As an initial step, we have included Figure A8, which illustrates how an input feature map for a specific channel is transformed through stimulus-specific affine operations.
>
>
> We are also actively exploring which aspects of language are most predictive of visual cortex activity, and how information may be shared across visual and linguistic modalities. As a preliminary analysis, we examined how different linguistic manipulations affect model performance (Table A7), including versions of sentences with only content words or with jumbled word order. Notably, the original, full sentences consistently yield the strongest or near-strongest performance, suggesting that holistic sentence-level representations provide a more robust foundation for modeling visual cortex responses than partial or reordered linguistic inputs.
>
>
> Lastly, we have also updated the main text with citations [1],[2] and [3] at lines 619-620.

---

> > ### Comment · Reviewer_jbu7 · 2025-04-17
> >
> > I appreciate the authors' responses and have updated my "interest" score accordingly. (Both of the other scores I had already "maxed out" and I'm admittedly not sure what the acceptance criteria are, but hopefully this helps!

---

> > > ### Author Response · Authors · 2025-04-21
> > >
> > > Thank you for the very helpful feedback (on this and previous iterations of our paper) and strong endorsement!

---

### Official Review · Reviewer_tkVP · 2025-03-31
**The manuscript introduces an effective and novel brain-alignment method, but the model comparisons provide limited new insights into image processing along the visual pathways.**

**Soundness:** 2
**Clarity:** 1

**Comments:**

There are various types of DNNs and brain-alignment methods that have successfully captured brain responses to images. In this manuscript, the authors compare how well brain responses are explained by model representations with different inputs (vision vs. language), optimizations (task vs. brain), and alignment methods (ridge regression, factorized ridge regression, voxel-wise Gaussian receptive field estimation, and STN, extended by stimulus- and feature-dependent receptive field estimation). After establishing that STN explains brain responses best across all brain regions and DNN versions, they show that early visual cortex aligns best with response-optimized network representations, while higher visual cortex aligns best with task-optimized network representations. Additionally, they find that semantic representations of high spatial resolution best explain activation in early visual cortex, and overall image semantics higher visual cortex.

The aim of the manuscript is twofold: First, it presents a valuable extension of state-of-the-art methods to link DNN representations to neural responses, utilizing the variability of receptive fields in light of specific stimuli and their features. Second, it raises the potentially interesting question of what features are represented along the ventral visual pathway. However, this dual agenda results in a very dense manuscript that insufficiently explains key terms and goals, and has an inconsistent structure that is difficult to follow. More critically, the broad range of feature spaces employed is evaluated solely based on their power to explain neural responses, thereby leaving open what characteristics of these feature spaces actually give rise to brain alignment. This severely limits the interpretability and impact of the findings. I believe this manuscript would benefit from being split into two separate papers, allowing the new brain-alignment technique to stand out more clearly rather than being overshadowed by the weaker experimental design of the model comparisons.

Major comments:
1. The study compares brain scores for a wide range of models but does not specify which characteristics of these models explain neural responses. Models vary along many dimensions, including the number of parameters, connectivity, objective functions, and input modalities. Consequently, an increase in brain score cannot be easily attributed to one specific characteristic of a model as intended by the authors. To address this, it would be necessary to either systematically manipulate model characteristics across all these dimensions or characterize what representations of a model exactly the brain picks up on. The study’s central claim about brain gradients from response- to task-optimized network representations, as well as from semantics of high to low spatial resolution, is therefore unconvincing.
2. Following up on the above point, could there be some basic model characteristics that explain observed effects of model inputs and optimization? For instance, does the dimensionality of winning model representations align best with the dimensionality of respective brain areas?
3. Throughout the manuscript, the main objective and structure are unclear. The introduction focuses on feature encoding along the visual pathway, but fails to present any hypotheses about underlying processes in the brain that would make these considerations relevant. Readouts appear more as a side note in the end. The methods section only distinguishes between encoders and readouts, with inputs becoming part of the encoder section, whereas sometimes they are considered an independent dimension. In the results section, readouts are presented first, with their results determining all comparisons carried out for the encoder and input comparisons. Finally, the discussion merely summarizes the results without addressing implications for underlying processes in the brain, making it difficult to identify the novelty of the study.

Minor comments:
1. The title needs a more unifying message that highlights the value of this work beyond the various model manipulations.
2. The abstract is difficult to follow. Many terms used are jargon-heavy and not self-explanatory. The different types of analysis aren't clearly distinguishable, and the relevance of the findings for advancing our understanding of the brain remains unclear.
3. The inputs dimension and its relevance are not adequately addressed in the introduction, despite being a main component of the study.
4. How is the quality of GPT-2-generated dense captions evaluated and ensured?
5. The newly developed readout method proposed by the authors is not sufficiently motivated in the introduction.
6. What makes the "Semantic Spatial Transformer Readout" semantic?
7. The labels in figures are too small.
8. It is not clear which input, encoding, and readout variants can be combined. Please illustrate this more clearly.

**Expertise:**

2

**Interest:**

2

---

> ### Author Rebuttal · Authors · 2025-04-15
>
> Thank you for the feedback. We offer the following clarifications:
>
> *What model characteristics drive brain alignment?*  Instead of isolating every possible factor, we focus on comparing 3 broad modeling paradigms—task‐optimized (TO) and response‐optimized (RO) vision, and language‐only models. We examined multiple TO vision models (ResNet, AlexNet, ConvNeXt, MaskRCNN, sup vs self-sup) and language models (MPNET, CLIP, GPT‐2), ensuring that results aren’t tied to a particular design choice. We consistently find early visual areas are best modeled by purely visual representations, while higher regions benefit from semantic representations.
>
> While architecture has little impact on TO performance (consistent with Conwell et al.), RO models show strong architectural sensitivity (Table A6). Those mimicking TO architectures underperform—due to training on only ~3% of the data used by TO models. In contrast, adding rotation-equivariance significantly improves performance, indicating that targeted inductive biases can help mitigate data limitations and enable RO models to approach TO accuracy.
>
> *Could dimensionality explain these effects?* As each voxel is modeled independently, # voxels in a region do not affect model fit. Moreover, the RO model uses a single architecture for all regions, providing a stable dimensionality baseline. We also validated that adding more TO or language‐model baselines yields the same modality‐specific patterns (see response to reviewers eya5/udya), minimizing the likelihood that trivial factors alone drive performance.
>
> *Revised Manuscript structure/goals* We first compare readout mechanisms—demonstrating their strong impact—then use the best readout to evaluate different encoders across the visual cortex. This ensures each encoder is fairly tested under its most effective readout. Our discussion now highlights the broader insight: explicit alignment with neural responses is crucial in the early visual cortex, while semantic or TO features better capture higher‐level regions.
>
> Minor Comments:
> Revised the abstract, intro and figures according to reviewer feedback.
>
> GPT‐2 captions primarily probe image‐driven vs. linguistic representations- usefulness is confirmed by strong empirical performance.
>
> “Semantic” in our proposed Readout refers to modulation of both feature maps and receptive fields based on the content of images.
>
> Single‐caption models use ridge regression only (Line 228-232); other inputs are compatible with all readouts.

---

> > ### Comment · Reviewer_tkVP · 2025-04-17
> >
> > I appreciate the authors' clarifications and revisions to the manuscript.
> >
> > The addition of more models to the comparisons indeed enhances confidence in the observed effects, and I have accordingly increased my score for soundness.
> >
> > The novelty of the results beyond the new method for brain alignment remains unclear to me. It has long been established that the visual processing gradient evolves from perceptual to semantic and categorical features. The results are not sufficiently embedded in the existing literature, making it difficult to discern their value.
> >
> > I understand that the time for revisions was very limited. However, claiming to incorporate reviewer feedback in the introduction, discussion, and figures without substantial changes (based on my comparison of the old and new manuscripts, which revealed only minor alterations to single words) seems misleading. Therefore, an important issue I raised—the manuscript's lack of structure and focus—still persists.
> >
> > Overall, while the study introduces an interesting new method for brain alignment, it still lacks a convincing presentation and does not clearly demonstrate the novelty and relevance of the brain alignment results.

---

> > > ### Author Response · Authors · 2025-04-21
> > >
> > > Thank you for your thoughtful review and for recognizing the value of our additional model comparisons. We appreciate your engagement with the manuscript and we want to further clarify our contributions-
> > >
> > > We would like to emphasize that the introduction of the Semantic Spatial Transformer readout represents a substantial methodological advance. This readout not only significantly outperforms existing baselines, but also enables a more nuanced understanding of the dynamic modulation of receptive fields in the brain — a direction with considerable potential for future research. Furthermore, we show that the choice of readout method markedly influences prediction accuracy—and, to our knowledge, no prior study modeling human brain responses has systematically benchmarked this full suite of readouts.
> > >
> > > Regarding the comment that the manuscript does not clearly demonstrate novelty beyond aligning with the well-established perceptual-to-semantic gradient in visual processing: our aim was not to challenge this principle, but rather to build upon it through a comprehensive and systematic comparison of model architectures and input modalities. These comparisons help pinpoint where and how different types of information (e.g., vision-only vs. language-informed —using single captions to capture global context versus dense captions for local detail) contribute to the modeling of distinct visual areas, which we believe offers meaningful insights into cortical organization.
> > >
> > > We acknowledge that our revisions to the manuscript were modest, particularly in terms of structural changes. This was due to the very limited time available for the revision. However, we will prioritize improving the clarity and focus of the presentation in the camera-ready version if the paper gets accepted.
> > >
> > > Once again, we thank the reviewer for their careful consideration and constructive feedback. We will continue to refine the manuscript to better convey both the novelty and the broader implications of our work.

---

### Meta-Review · Area_Chair_zqWv · 2025-04-30

**Ccn Recommendation:** Accept as Proceedings

**Metareview:**

The authors have addressed many of the reviewers’ concerns, and despite some limitations, most recommend acceptance as Proceedings. I agree with the reviewer's evaluations of the strengths and remaining weaknesses, particularly the extent to which it offers new neuroscientific insights beyond performance gains, despite its methodological innovations. I suggest integrating key appendix material into the main text and sharpening the clarity and focus of presentation, as the authors have promised. With confidence that these improvements will be implemented in the camera-ready version, I recommend acceptance as Proceedings.

**Summary:**

Reviewers highlighted the main contribution of this submission as the novel Semantic Spatial Transformer Readout, praising its innovative elements, which go beyond overly recycled, standard DNN approaches to address multiple methodological challenges in modelling the human visual cortex. They noted remaining weaknesses, including the extent to which the work yields new neuroscientific insights beyond performance gains, the need for deeper analysis of the specific model characteristics driving performance, and initial limitations in vision model choices (issues that the authors convincingly addressed by adding more complex vision baselines and a additional table and a figure in their appendix). However, one reviewer remains unconvinced and argues that the manuscript could better isolate model characteristics and articulate hypotheses to move beyond the well-known perceptual–semantic gradient of the visual pathway. Additionally, the technical sections were described as dense and difficult to follow, suggesting that further clarity improvements are needed in the camera-ready version. Overall, reviewers agreed on broad interest and adequate-to-strong soundness, with clarity ratings varying from “needs improvement” to “exceptional.” At the end of the review process, most reviewers recommend acceptance as a Proceedings, and this work makes a significant contribution to the CCN.

**Expertise:**

2